# Diversity, recombination and misclassification in the family *Geminiviridae*: Insight from bioinformatics analysis

Moshood Olamide Lateef[1,2], Dauda Nathaniel[3], Bolaji Osundahunsi[4],
Neil Arvin Bretana[1]*

**1** IU International University of Applied Sciences, Erfurt, Germany, **2** Virology and Molecular Diagnostics Unit, International Institute of Tropical Agriculture, Ibadan, Nigeria, **3** Department of Crop Sciences, Faculty of Agriculture, University of Nigeria, Nsukka, Nigeria, **4** Department of Entomology and Plant Pathology, University of Arkansas System Division of Agriculture, Fayetteville, Arkansas, United States of America

* neil-arvin.bretana@iu.org

## Abstract

The family *Geminiviridae* comprises 15 genera and over 500 species of diverse plant-infecting viruses. Previous studies on this virus's diversity primarily focused on the members' selection without considering the entire genomic divergence. This study presents comprehensive comparative genomic analyses and a global distribution map using 17,718 complete genomes and 28,185 geminivirus-associated metadata records. Intergeneric pairwise identity among monopartite and *Begomovirus* DNA-A is ≥40%, while interspecies identity among *Begomovirus* DNA-B is ≥41%, corresponding to genetic variations of ≤60% and ≤59%, respectively. *Begomovirus* and *Mastrevirus* are the only genera detected across all continents, with Asia exhibiting the highest genus diversity (11 genera). However, in Africa, geminiviruses from six genera are more broadly distributed across individual countries. To our knowledge, this is the first geographical map constructed using all genera from the *Geminiviridae* family. Phylogenetic reconstructions of complete genomes, coat proteins, and replication-associated proteins (Rep) underscored distinct clustering patterns consistent with genus-level classification but revealed exceptions, including misclassified viral species and potential new taxa. Notably, French bean severe leaf curl viru*s* with accession KC699544 was reclassified from *Capulavirus* to *Begomovirus* with the proposed name Corchorus yellow vein mosaic virus. Additionally, two unclassified begomovirus-like viruses, named "begomovirus spathoglottis 1" and "2," are proposed for reclassification within *Maldovirus* under the tentative names "maldovirus spathoglottis 1" and "2." The *Begomovirus* Rep showed phylogenetic affinity with Rep from *Maldovirus, Curtovirus, Turncurtovirus,* and *Topocuvirus,* supporting potential evolutionary relationships among these genera. Recombination analyses confirmed high-frequency interspecies and intergeneric recombination events, predominantly

**Data availability statement:** All relevant data are within the manuscript and its Supporting information files.

**Funding:** The author(s) received no specific funding for this work.

**Competing interests:** The authors have declared that no competing interests exist.

involving *Begomovirus*, underscoring its pivotal role in geminivirus evolution. Furthermore, we hypothesized specific vector transmission: *Opunvirus*, *Welwivirus*, and *Topilevirus* are transmitted via treehopper species, whereas *Citlodavirus* and *Eragrovirus* are transmitted via leafhopper species. Accurate identification and classification of plant viruses and their transmission vectors are essential for developing effective surveillance and management strategies to safeguard global agriculture.

## Introduction

*Geminiviridae* is a cosmopolitan virus family that causes several plant diseases and is transmitted by insect vectors [1,2]. It exists as monopartite or bipartite twinned virions that encapsulate the circular single-stranded deoxyribonucleic acid (DNA) [3,4]. Economically, these viruses cause huge losses to various plant hosts, such as vegetables, fiber, and root crops, mainly in the subtropical and tropical regions of the globe [5,6]. Several studies have reported severe crop losses caused by specific geminivirus species. Notably, cotton leaf curl disease (CLCuD), primarily caused by *Begomovirus gossypimultanense* and *Begomovirus gossypikokranense* in Pakistan and India [7], has hindered cotton production in these countries, while African cassava mosaic virus (ACMV, *Begomovirus manihotis*) is responsible for cassava mosaic disease across Africa [9]. In the United States, tomato yellow leaf curl virus (TYLCV, *Begomovirus coheni*) has significantly affected tomato crops [5]. The cotton industry in Pakistan alone suffers estimated annual losses of $5 billion due to CLCuD. Similarly, TYLCV causes approximately $140 million in annual losses to tomato production in the USA. Hema *et al*. [8] reported over $300 million in losses to the Indian bean industry attributed to begomoviruses. Additionally, cassava mosaic disease results in an estimated $2 billion in annual losses across Africa [9]. Currently, fifteen genera and over five hundred species have been identified as members of the *Geminiviridae* [10], making it the largest family of plant viruses. The genera include *Capulavirus, Citlodavirus, Curtovirus, Eragrovirus, Grablovirus, Maldovirus, Mastrevirus, Mulcrilevirus, Opunvirus, Topilevirus, Topocuvirus, Turncurtovirus, Welwivirus, Becurtovirus,* and *Begomovirus*. Among these genera, *Begomovirus* contains over four hundred species and remains the most studied. New genera and species are continually being detected due to advances in sequencing technology with the use of next-generation sequencing to detect new and unclassified viruses [11–13].

Insect vectors are the major mode of spread of these viruses, with a single vector capable of transmitting viruses to multiple species or genera of susceptible hosts. This remains a potential source for the intra- or inter-genus recombinant virus. However, mutation, recombination, and pseudo-recombination are the driving forces of geminivirus evolution [14]. The emergence of new virus species and genera with modified virulence can overcome host resistance genes and challenge surveillance due to the virus's increased adaptation to environmental factors and susceptible host plants [15–17]. This menace potentially challenges food security across the world.

Recombination has been the driving force behind the evolution and emergence of new viruses within the *Geminiviridae* family through possible adaptation to changing environmental conditions and susceptible host plants [15]. Several studies have reported intra- and interspecies recombination leading to the emergence of new viruses within *Geminiviridae* [18–22]. These studies mainly focused on begomoviruses and a few other genera. Therefore, all 15 genera of *Geminiviridae* need to be considered for recombination analysis to provide a comprehensive view of recombination across the entire family. This study's sampling approach considered every virus genome with either high or low sequence diversity to represent the full sequence diversity of this highly diverse virus family.

The distribution of all geminiviruses is reported to be global [23–25]; however, these studies depicted the global distribution through a geographical map with data points on a few genera, whitefly vector distribution, or geminivirus-based satellites. A comprehensive map showing the global distribution of all genera is still lacking. The construction of such a map will provide substantial information at a glance for prompt decision-making. Understanding the diversity and evolution of both new and previously reported species is essential to support surveillance efforts against emerging diseases threatening food security. This study therefore focuses on the patterns of diversity and recombination within and among all 15 genera and a geographical distribution map for the entire genera*.*

## Materials and methods

### Sampling procedure and techniques

The complete genomes of all available monopartite and bipartite viruses of the family *Geminiviridae* available until May 10, 2024, were retrieved from the GenBank [26]. This included 14,778 genomes of monopartite viruses and DNA-A components of bipartite viruses and 2,940 genomes of DNA-B of bipartite viruses. A chart showing the number of retrieved and sampled sequences across each genus is included in the supplementary materials (S1 and S2 Figs). The downloaded sequences were exported to the sequence demarcation tool (SDT) version 1.3, a tool that classifies sequences based on their percentage pairwise identity [27]. The sequences with the same percentage identities were grouped together and served as a pool to select representative samples. The genetic diversity among these genera varies. Conventionally, geminivirus sequences with ≤ 95% pairwise identity were chosen for the study performed by Padidam *et al.* [21]. In this study, ≥ 95% pairwise similarity in viral sequences within each of the *Topocuvirus, Topilevirus, Maldovirus,* and *Eragrovirus* was partitioned, and sequence(s) were sampled from each partition using SDT. Similarly, ≥ 92% pairwise similarity was used for *Capulavirus, Citlodavirus, Curtovirus, Becurtovirus, Grablovirus, Mulcrilevirus, Opunvirus, Turncurtovirus,* and *Welwivirus.*

Sequences for *Mastrevirus* and *Begomovirus*, which had a large number of available sequences, were grouped into batches for parallel computation, and sampling was performed using ≥80% pairwise similarity. Similarly, Bandoo *et al*. [28] adopted 80% pairwise identity when selecting members of *Begomovirus*. The detailed procedures for virus genome sampling and geographical map construction are available at https://doi.org/10.17504/protocols.io.j8nlkybjdg5r/v1. The selected sequences from each genus partition were gathered and used as representatives for the respective genus. In addition, the corresponding amino acid sequences of the coat protein (CP) and replication-associated protein (Rep) for all the sampled nucleotide sequences across all genera were retrieved from the GenBank. Details of these sampled geminiviruses are provided in the supplementary materials (S1 Table).

### Intra-geminivirus sequence alignment

The sampled virus sequences for each genus were imported into individual FASTA files and aligned with multiple alignments using Fast Fourier Transform (MAFFT) algorithm version 7.526 [29]. The options L-INS-I and --ep were selected to efficiently align the sequences. The alignment was performed with and without outgroup sequences for the diversity and recombination analysis, respectively. The chicken anemia virus with accession number M55918.1 was used as the

outgroup for the nucleotide-based phylogenies, consistent with previous studies [3,30]. BioEdit software version 7.2.5 [31] was used to view and monitor the alignments manually. A few sequences with incorrect orientation were corrected, and the sequences were realigned.

### Distribution across the world

The metadata for all geminiviruses, including both complete and partial genome sequences, were also retrieved from GenBank and served as the main data for the map illustration. Explicitly, the species of the virus and the location source were aggregated by their respective genera and source countries to construct the geographical map (S2 Table). This approach minimizes graphic clutter and facilitates data visualization. The Plotly library in Python programming was used to construct the distribution map.

### Phylogenetic-based evolutionary relationship

The evolutionary relationships among all genera of the *Geminiviridae* were constructed with the phylogenetic tree. For all genera, as well as for the DNA-A of bipartite begomoviruses, aligned nucleotide sequences of complete genomes and amino acid sequences of Rep and CP were used to reconstruct phylogenetic trees using IQ-TREE [32]. The best-fitting substitution model was automatically selected based on Bayesian Information Criteria with 1,000 ultrafast bootstrap replicates.

In contrast to the nucleotide-based phylogeny, the amino acid-based phylogenies were constructed without an outgroup and were instead rooted at the midpoint. The reconstructed phylogenies were edited and visualized using FigTree version 1.4.4 [33].

### Recombination analysis

The multiple sequence alignments without an outgroup were imported into the recombination detection program version 5 (RDP5) [34]. The following integrated methods in RDP5 were used to perform a preliminary scan for possible recombination in the aligned virus sequences: Chimaera [35], Bootscan [36–37], Geneconv [21], SisterScan [38], Maximum Chi Square [35,39], and 3Seq [40]. A Bonferroni-corrected P-value of 0.05 and default parameter settings were used for the individual method analysis. To increase the stringency and reduce false positives, recombinants that were identified by at least four methods were considered reliable.

## Results

### Genetic variation and pairwise identity across *Geminiviridae*

Analysis of pairwise identity for the sampled genomes revealed that inter-genera identity among monopartite geminiviruses and *Begomovirus* DNA-A is ≥40% (Fig 1), corresponding to a maximum genetic variation of 60%. Similarly, interspecies pairwise identity across the complete genomes of *Begomovirus* DNA-B is ≥41% (Fig 2), corresponding to a maximum genetic variation of 59%.

### Mapping the global distribution of *Geminiviridae*: Continental spread of the viral genera

The geographical distribution map (Fig 3) constructed with the metadata of 28,185 data points (metadata for both complete and partial genomes) revealed that members of the family *Geminiviridae* are globally distributed across all six continents. However, only the genera *Begomovirus* and *Mastrevirus* have been isolated from every continent. Eleven out of the fifteen genera within *Geminiviridae* were isolated from Asia. These genera are *Begomovirus, Mastrevirus, Curtovirus, Citlodavirus, Grablovirus, Topocuvirus, Turncurtovirus, Becurtovirus, Capulavirus, Maldovirus,* and *Mulcrilevirus*. South America is the second continent with seven genera reported, namely *Begomovirus, Mastrevirus, Citlodavirus, Grablovirus, Topilevirus,*

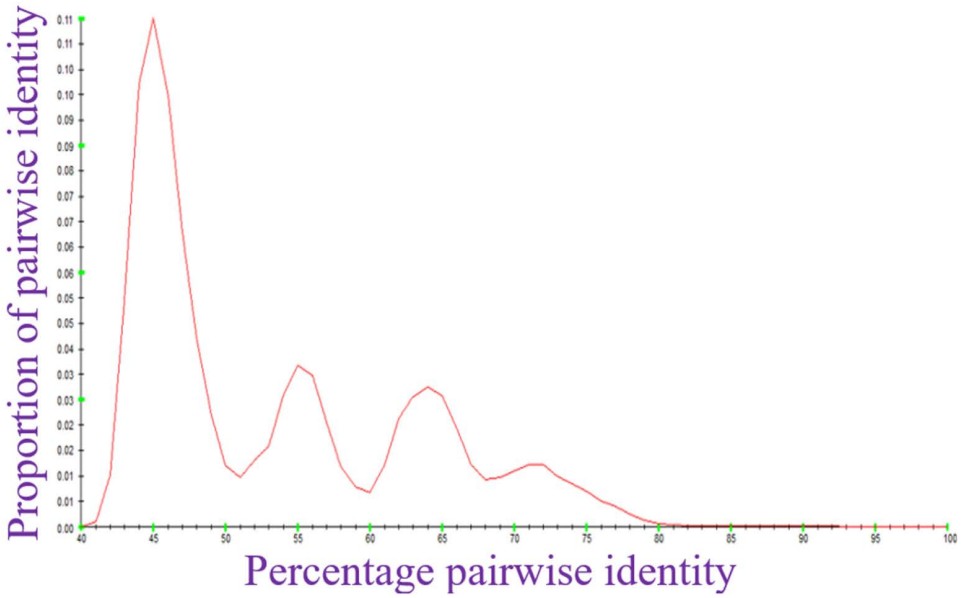

**Fig 1. Pairwise identity of the sampled monopartite and DNA-A of *Begomovirus* genomes using SDT.**

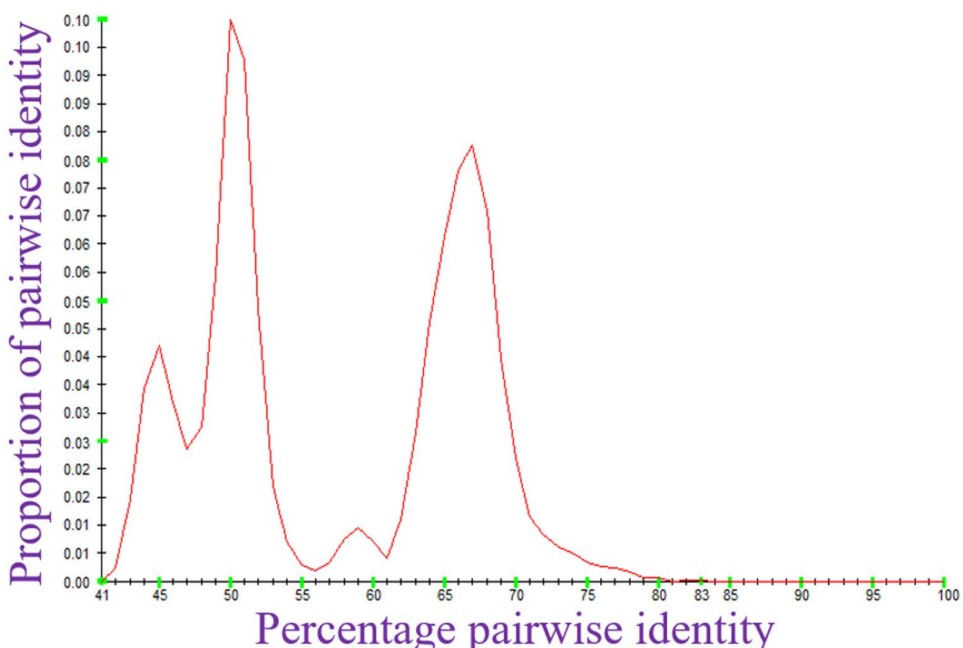

**Fig 2. Pairwise identity of the sampled DNA-B of *Begomovirus* using SDT.**

*Capulavirus,* and *Welwivirus*. North America and Africa have cases of six genera reported from each of the continents. *Begomovirus, Mastrevirus, Grablovirus,* and *Becurtovirus* were reported from both continents. However, *Curtovirus* and *Opunvirus* were reported from North America, while *Capulavirus* and *Eragrovirus* were the additional reported genera from Africa.

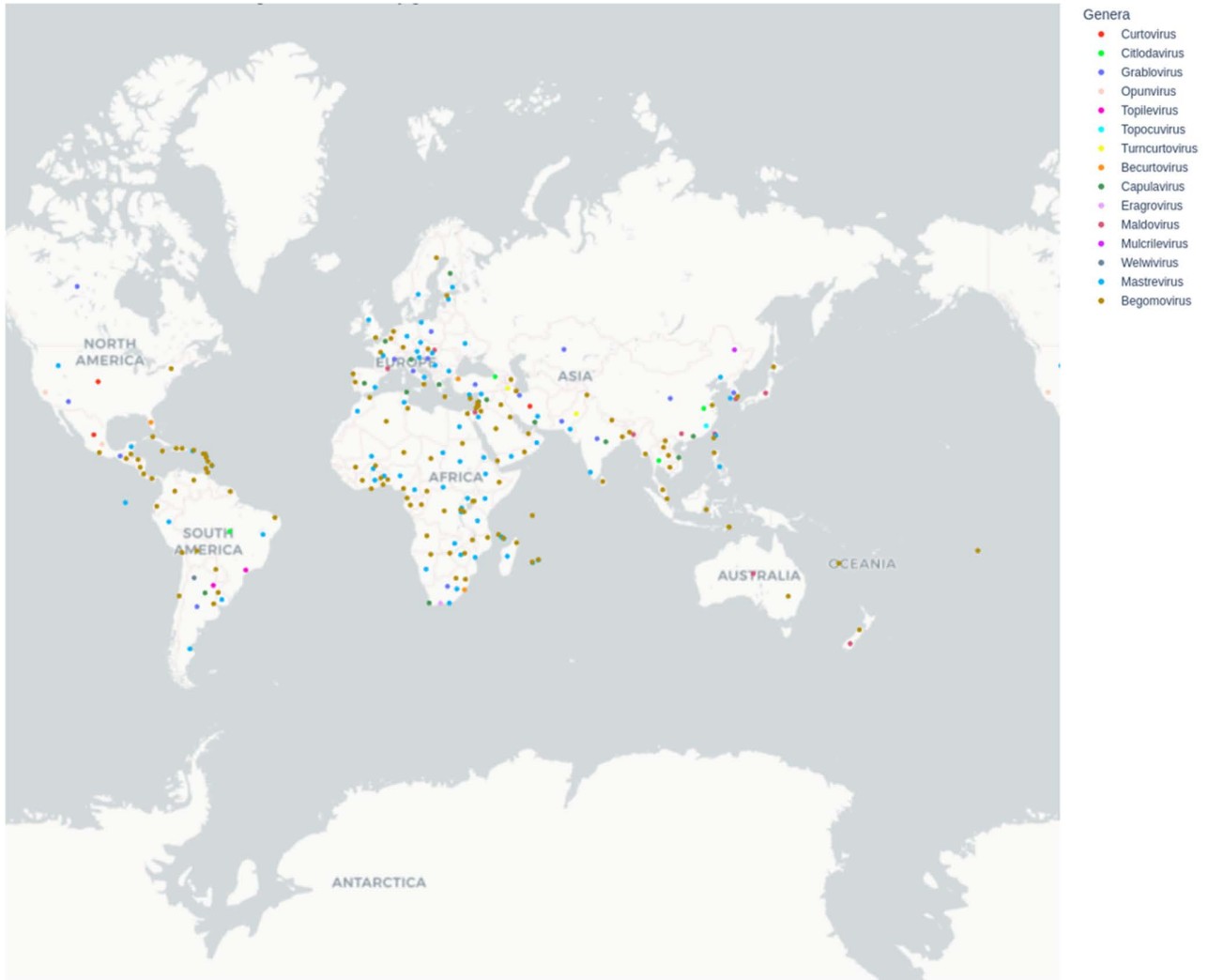

**Fig 3. Geographical map showing global distribution of genera within *Geminiviridae*.** Constructed with Plotly library.

Additionally, the African continent had the highest number of countries where geminivirus infections are reported. Europe has five genera of *Geminiviridae* reported within the continent. These are *Begomovirus*, *Mastrevirus*, *Grablovirus*, *Capulavirus,* and *Maldovirus*. Only three genera—*Begomovirus*, *Mastrevirus*, and *Maldovirus*—have been reported from Oceania.

**Phylogenetic patterns in *Geminiviridae***

The total number of sampled sequences used to construct the phylogenetic trees was 284 for the complete genome, 280 for the coat protein, and 282 for the Rep protein. After multiple sequence alignments, the orientation of all sequences was checked and confirmed. GTR+F+R9, LG+F+I+G4, and rtREV+F+G4 were the best-fit chosen models, respectively, for the nucleotides, CP, and Rep of monopartite and *Begomovirus* DNA-A. The reconstructed phylogeny with the virus genomes (Fig 4) revealed the members of every genus to form a cluster, subsequently forming a clade between genera. However, a few *Begomovirus* species, including unclassified *Begomovirus*, cluster outside their respective genera. Similarly, the CP of every genus forms a clade, with the exception of a few unclassified begomoviruses (Fig 5). Contrarily, the

 

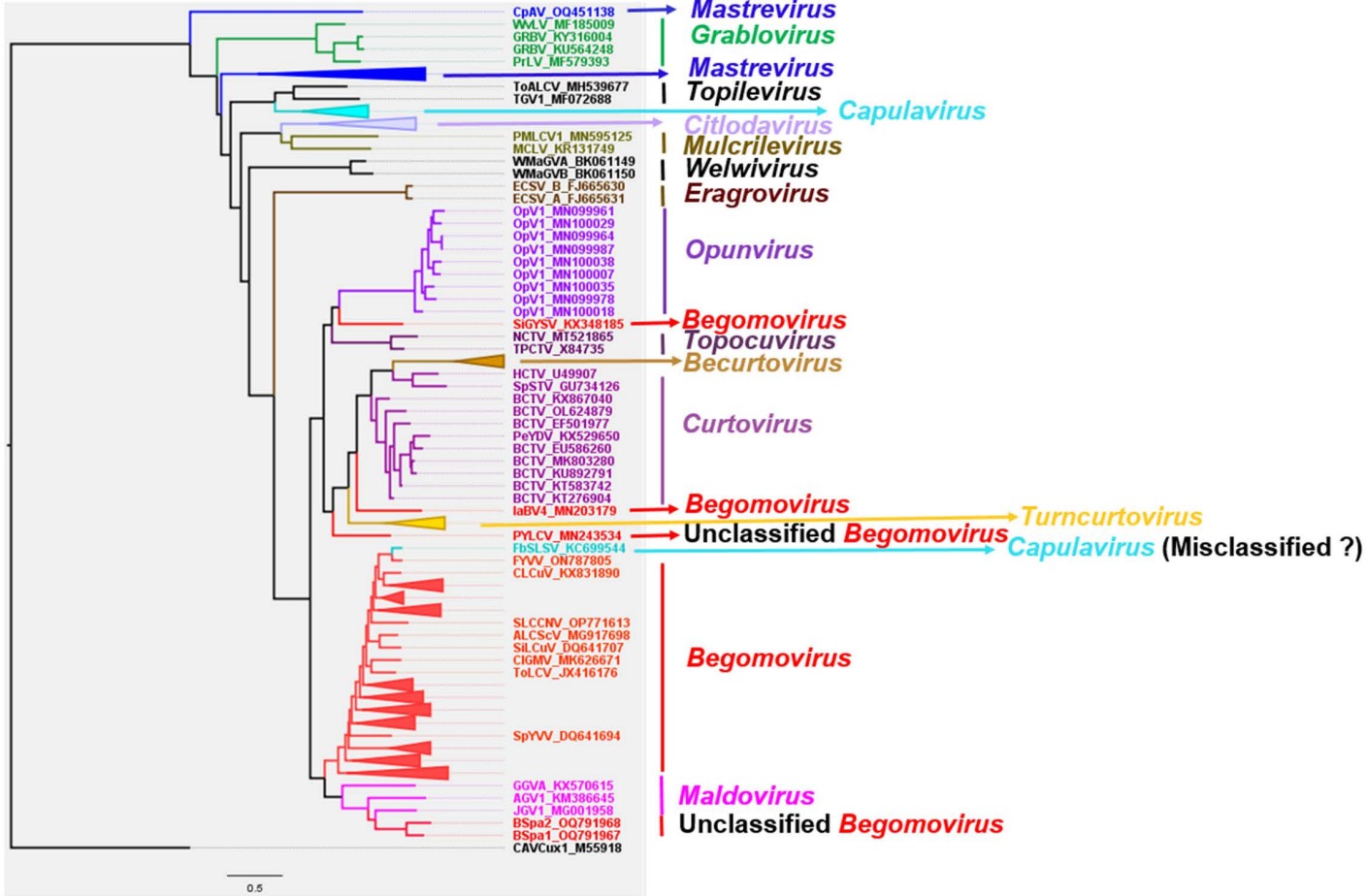

**Fig 4. Phylogenetic reconstruction with the nucleotides of monopartite and the DNA-A of the *Begomovirus* genome.** Inferred using maximum likelihood method within IQ-tree software at 1000 ultrafast bootstrap replicates.

Rep of the virus species within the genus *Begomovirus* forms many clusters that are distributed across the Rep of other genera, including *Curtovirus, Maldovirus, Turncurtovirus,* and *Topocuvirus* ([Fig 6]).

### Identifying and analyzing a misclassified viral accession

While analyzing the result of the phylogenetic reconstruction, French bean severe leaf curl virus (FbSLCV) with accession KC699544, classified as a member of the genus *Capulavirus,* formed a clade with the *Begomovirus* genome, CP, and Rep sequences in their respective phylogenies ([Figs 4]–[6]). However, it is possible for one or a few of all the molecular compositions (genome, CP, and Rep) of geminivirus to be more closely related to a particular different genus, but not for the entire composition. This prior knowledge prompted further analysis to complement the reconstructed phylogenies. The complete genome nucleotide and the amino acid sequences for the full CP and Rep of this virus accession were independently analyzed using BLASTN and BLASTP against the GenBank database; the result revealed several virus members of *Begomovirus*. Interestingly, no *Capulavirus* species were detected in the BLAST results.

The entire sample of representative viruses for the genus *Capulavirus* and some of the closest viruses from the BLAST result were subjected to the pairwise identity analysis using SDT version 1.3 [27]. ClustalW within the SDT tool was

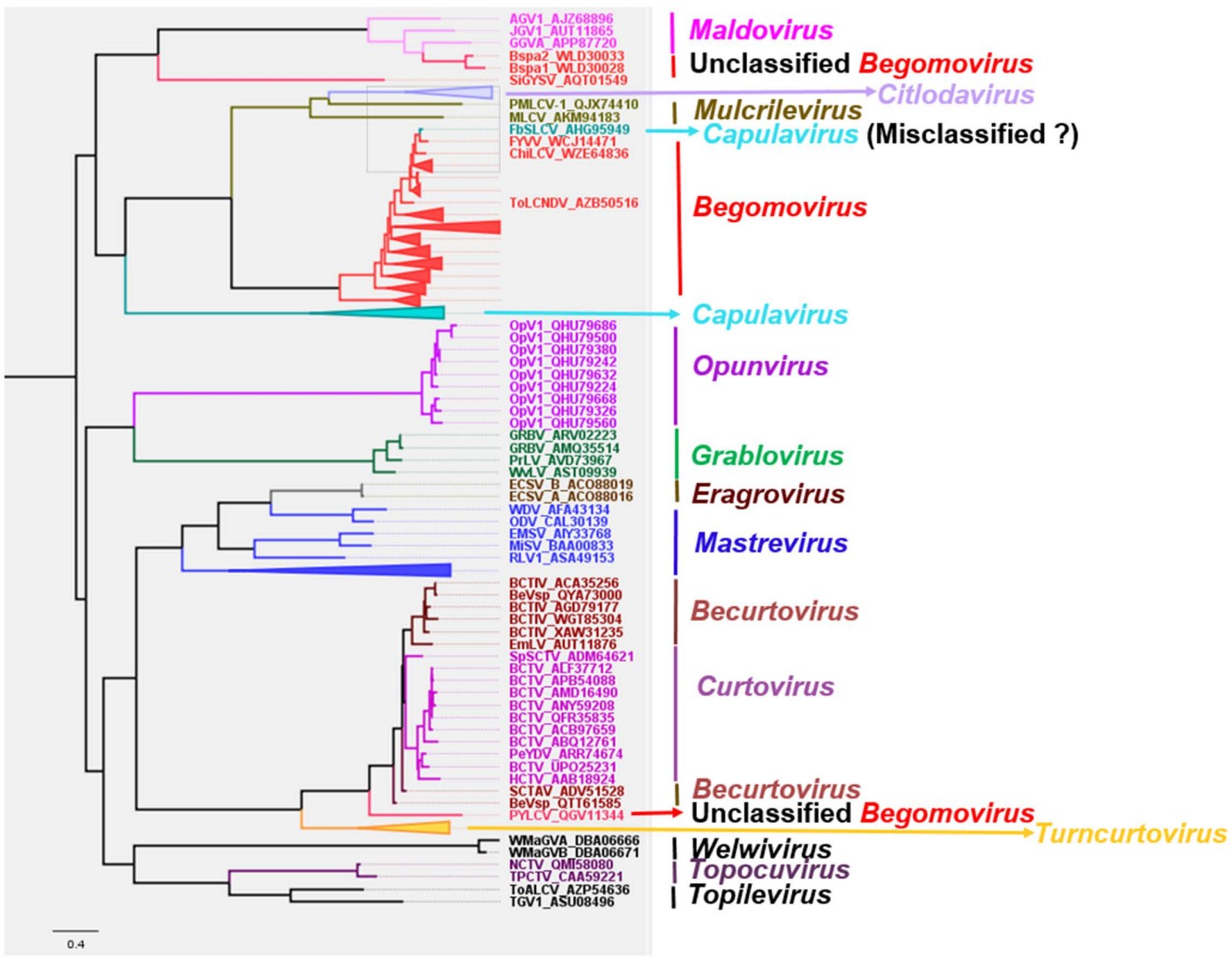

**Fig 5. Phylogenetic reconstruction with the coat protein of monopartite and the DNA-A of the *Begomovirus*.** Inferred using maximum likelihood method within IQ-tree software at 1000 ultrafast bootstrap replicates.

selected to align the sequences. The result showed the virus to have a close pairwise identity to the selected *Begomovirus,* while the disparity is too distant from the compared *Capulavirus* (Figs 7–9). To explicitly visualize the pairwise identity of this misclassified virus sequence (KC699544) to each of the selected viruses, the percentage similarity scores were extracted and used to plot a multiple bar chart for the complete genome, CP, and Rep regions, as shown in Fig 10.

The virus nucleotide sequences and amino acid sequences of the coat protein have the highest percentage pairwise identity of 91.3% and 99.2% to Corchorus yellow vein mosaic virus (CoYV) accessions KX513862 and AGG18212, respectively. Similarly, the virus Rep was found to be closely related to the French bean leaf curl virus (FbLCV) with a pairwise identity of 92.3% with accession number AFM77725. All these closely related viruses belong to the genus *Begomovirus*. Summarily, the virus is more closely related to CoYV of the genus *Begomovirus,* suggesting a new isolate. Hence, the accession is proposed to be named as Corchorus yellow vein mosaic virus within the genus *Begomovirus*.

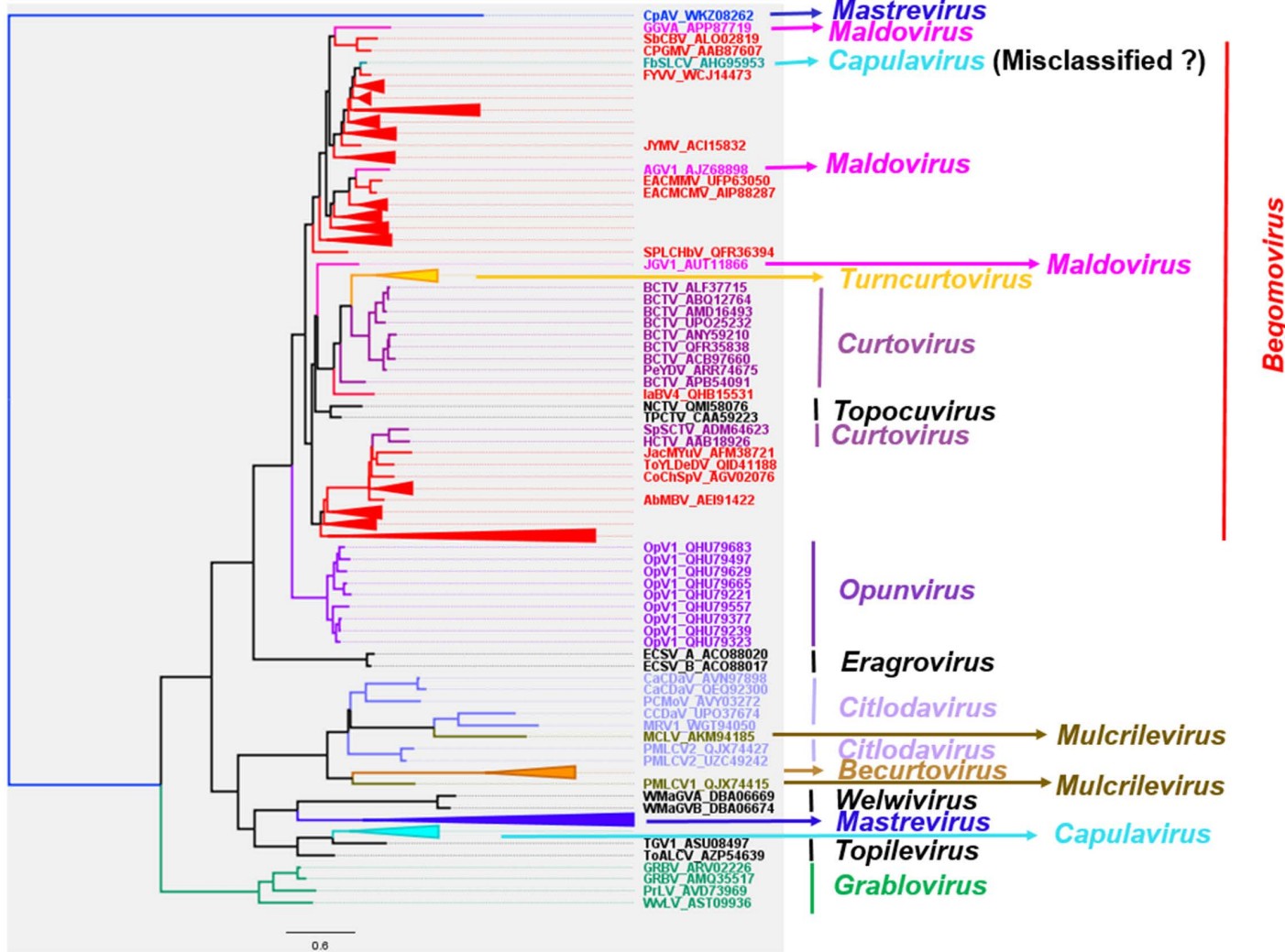

**Fig 6. Phylogenetic reconstruction with the Rep protein of monopartite and the DNA-A of the *Begomovirus*.** Inferred using maximum likelihood method within IQ-tree software at 1000 ultrafast bootstrap replicates.

## Proposed reassignment of unclassified viruses to the genus *Maldovirus*

Two unclassified begomovirus-like viruses, tentatively named "begomovirus spathoglottis 1" and "begomovirus spatho-glottis 2," were found to cluster phylogenetically with members of the genus *Maldovirus* (Figs 4 and 5). Notably, the clade containing these two viruses forms a sister group to the clade comprising established *Begomovirus* species, prompting further investigation through pairwise identity analysis (Fig 4). To assess genomic similarity, "begomovirus spathoglottis 1" and "2" were queried against the GenBank reference database using BLASTN. Genome sequences (full genome nucle-otide sequences, Rep, and CP) of a few top-hit viruses were retrieved, along with representative members of the genus *Maldovirus*, for comparative analysis. Pairwise sequence identity was calculated using SDT with default parameters.

Identity scores between apple geminivirus 1 (a member of *Maldovirus*) and the selected viruses were plotted (Fig 11). The genome-wide nucleotide identity across all viruses analyzed ranged from 62% to 65%. The CP was identified as the primary genomic region distinguishing *Begomovirus* species from both *Maldovirus* members and "begomovirus spathoglottis 1" and

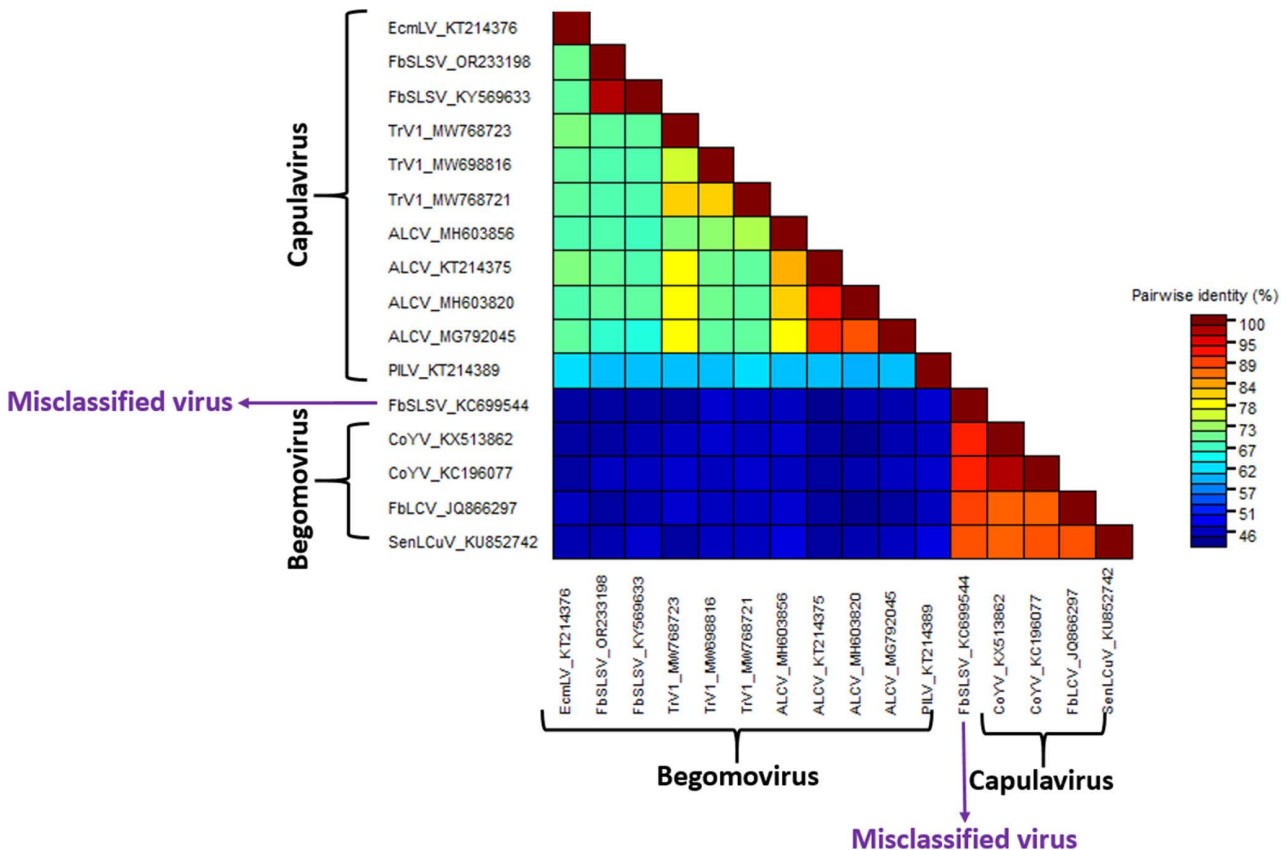

**Fig 7. Pairwise identity of the misclassified virus genome.**

"2" (Fig 11). The CP of *Maldovirus* members and "begomovirus spathoglottis 1" and "2" shared 45–48% pairwise identity compared to 18–22% with the selected begomoviruses. These results support the phylogenetic analyses (Figs 4 and 5), which indicate a close relationship between *Maldovirus* and these two unclassified viruses. Based on these findings, we propose that "begomovirus spathoglottis 1" and "begomovirus spathoglottis 2" be reclassified within the genus *Maldovirus*, under the tentative names "maldovirus spathoglottis 1" and "maldovirus spathoglottis 2." It is important to note that these proposed virus names remain provisional and are subject to formal approval by the International Committee on Taxonomy of Viruses (ICTV).

### Recombination-driven evolution in *Geminiviridae*

Interspecies recombination was more frequent in *Begomovirus*, with many species containing recombinant sequences derived from other species. Additionally, interspecies recombination was also detected in various species of *Begomovirus* DNA-B. Tables detailing the recombinant viruses are included (S3–S5 Tables). Additionally, inter-genera recombination was detected in *Opunvirus, Curtovirus, and Topocuvirus.*

### Discussion

The sampling approach adopted in this study considers the entire diversity in all the complete genome sequences in the family *Geminiviridae*. To the best of our knowledge, this is the first constructed geographical map using the entire genera

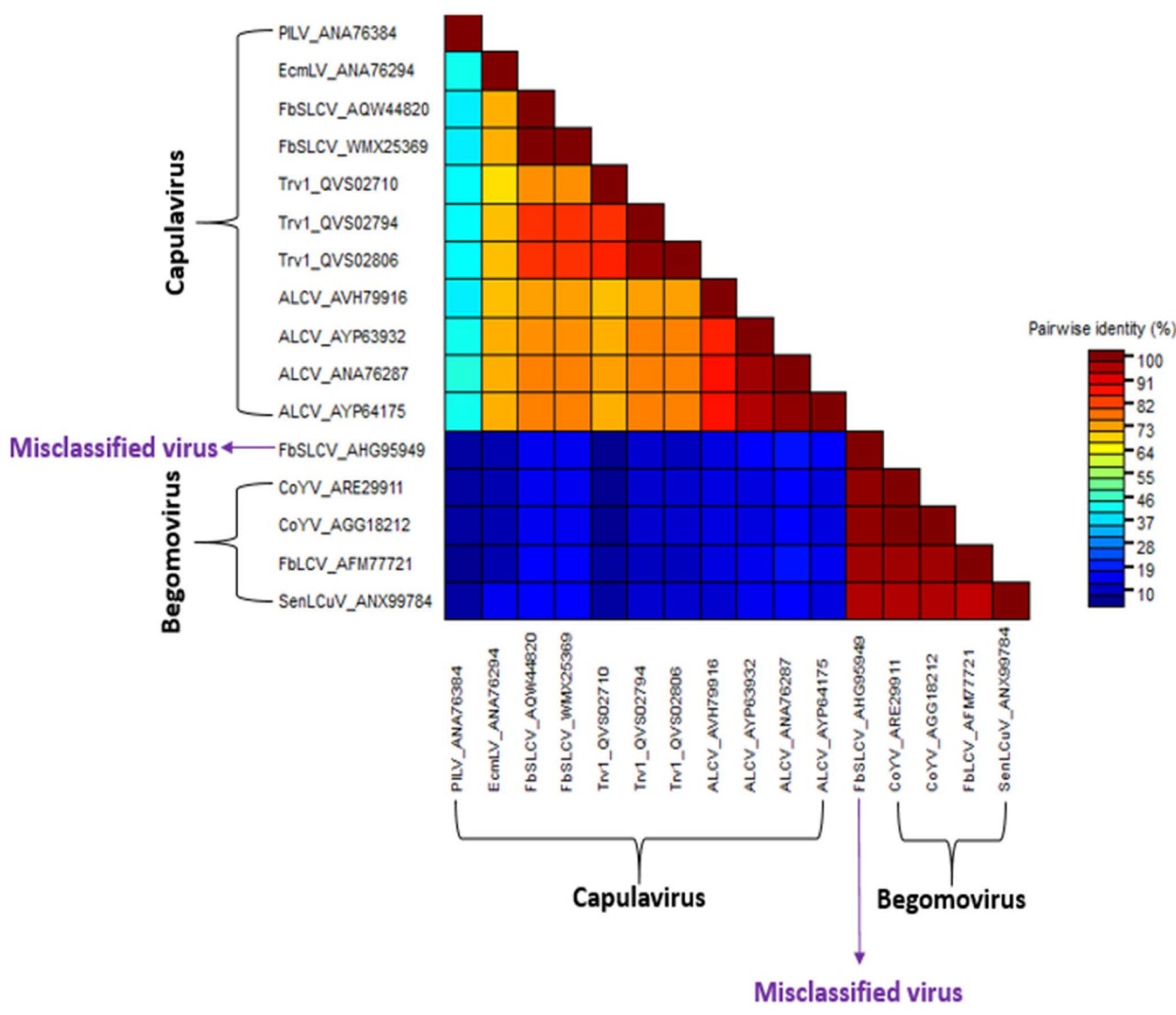

**Fig 8. Pairwise identity of the misclassified virus coat protein.**

of the family *Geminiviridae*. Previous studies used a few genera, such as *Begomovirus* [23], whitefly-infecting *Begomovirus* [25], and weed-infecting geminivirus [24], to depict the global distribution of geminivirus.

Notably, *Begomovirus* and *Mastrevirus* were the only genera found across all continents, a pattern consistent with previous studies highlighting their broad host range and efficient transmission by insect vectors such as whiteflies (*Bemisia tabaci*) and leafhoppers (*Cicadellidae*) [41,42]. The highest diversity of geminivirus genera exhibited by Asia may be attributed to the region's rich agroecological zones, dense cultivation of susceptible crops, and favorable climates that support vector populations.

South America ranked second in diversity, with seven genera reported. While the region showed lower genera diversity than Asia, the presence of less commonly detected genera such as *Topilevirus* [43] may suggest localized evolution or limited dispersal beyond the region. Africa stood out for having the highest number of countries affected, emphasizing its significance as both a center of geminivirus diversity and vulnerability [44]. This finding

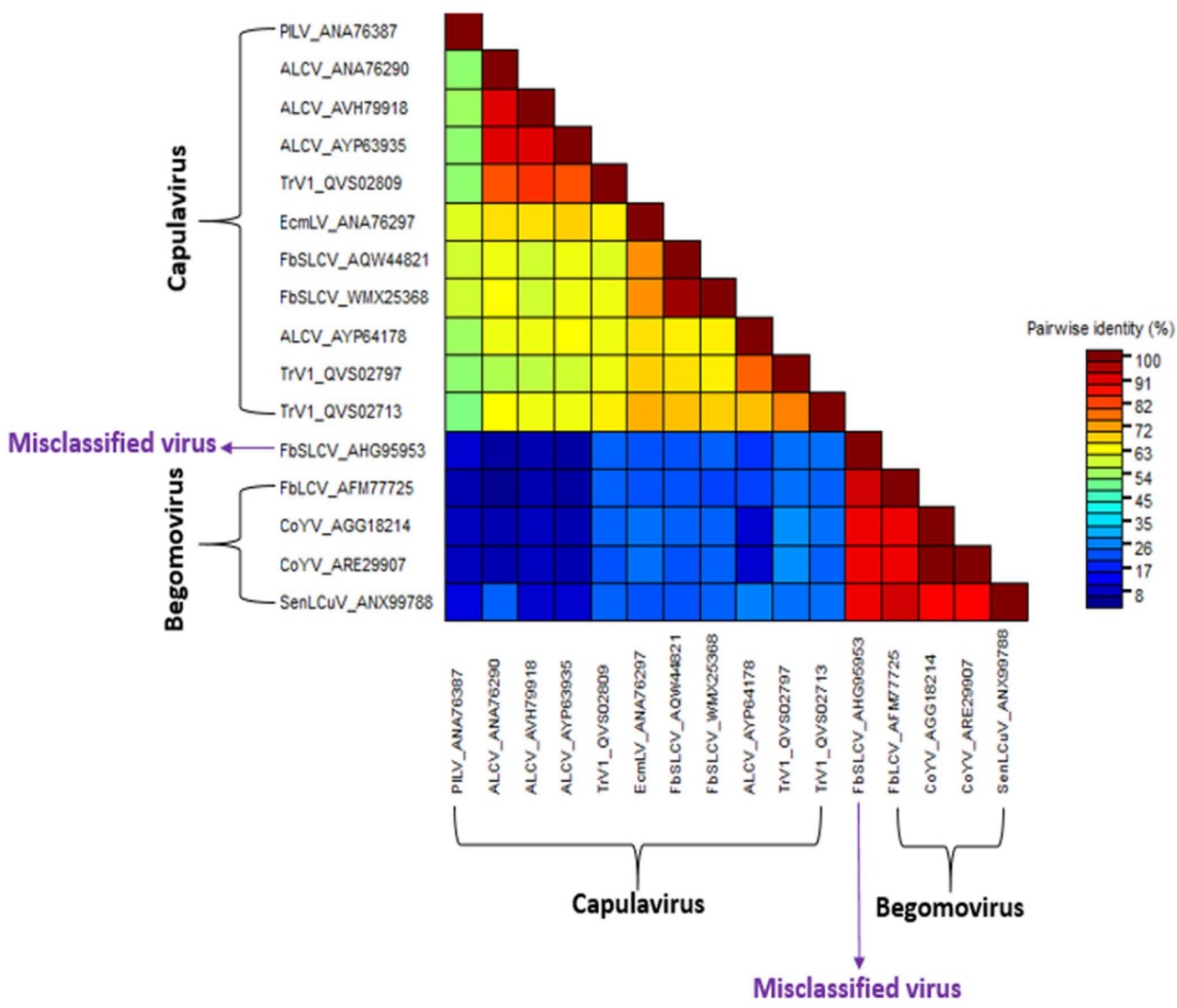

**Fig 9. Pairwise identity of the misclassified virus replication-associated protein.**

may be driven by widespread vector presence, the predominance of smallholder farming systems, and limited phytosanitary infrastructure. The lower genera diversity in Europe and Oceania could be a true reflection of limited viral presence or could result from lower sampling intensity and fewer molecular diagnostics being performed. Nevertheless, the presence of *Begomovirus* and *Mastrevirus* across all regions supports their dominant role in global geminivirus spread and evolution.

Summarily, while *Geminiviridae* members are globally distributed, their occurrence varies by region. This uneven distribution is likely influenced by multiple factors, including vector distribution, agricultural practices, plant host range, and environmental conditions, as well as the intensity of viral surveillance and reporting systems [45]. These findings underscore the need for enhanced global and regional surveillance, particularly in regions with high reported diversity, such as Africa, Asia, and South America. Understanding the evolutionary dynamics and ecological niches of these viruses remains crucial for developing effective management strategies.

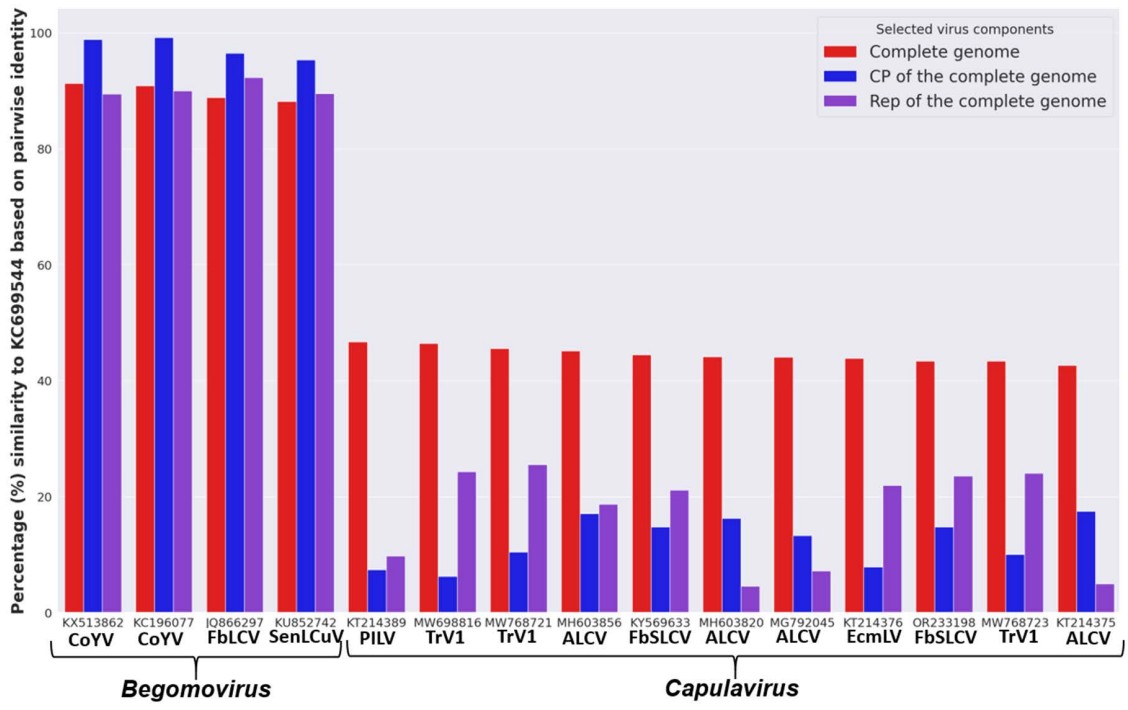

**Fig 10. Comparison of Pairwise identity of the misclassified virus to *Begomovirus* and *Capulavirus*.**

The ≥40% inter-genera identity across the family supports the idea of a shared evolutionary origin, despite the extensive diversification driven by recombination, host adaptation, and vector specificity [19,46]. Moreover, the DNA-B segment with ≥41% interspecies identity exhibits less recombination-driven divergence due to its limited gene content and essential movement functions [47]. These pairwise identity calculations affirm the robustness of current molecular criteria used for virus classification in the family *Geminiviridae.*

The comprehensive phylogenetic and comparative genomic analyses conducted in this study offer important insights into the evolutionary biology, host adaptation, and vector transmission strategies of members of the *Geminiviridae* family. This study reveals the *Citlodavirus* genomes and CP are closely related to *Mulcrilevirus,* while the Rep is similar to the Rep of virus members within *Mulcrilevirus* and *Becurtovirus*, consistent with the phylogenies inferred by Roumagnac *et al.* [43]. Similarly, the *Curtovirus* genome and CP are closely related to *Becurtovirus*. While the *Curtovirus* Rep clusters with the *Turncurtovirus* Rep, the *Becurtovirus* Rep clusters with members of *Mulcrilevirus*. The cluster of *Curtovirus* and *Becurtovirus* genomes and CP form a clade with the respective components of *Turncurtovirus,* consistent with the reported finding of Varsani *et al.* [48]. The close phylogenetic relationship of *Curtovirus* and *Becurtovirus* to *Turncurtovirus* supports the hypothesis that these genera share a common evolutionary origin [48], potentially shaped by overlapping host ranges or vector interactions. The genome sequences and the Rep amino acids of the *Maldovirus* are closely related to *Begomovirus* in this study; this is consistent with the inferred relationship by Roumagnac *et al.* [43].

*Opunvirus* (OpV1) genomes form a clade with genus *Topocuvirus*. However, Fontenele *et al.* [49] reported this genus to be close to *Topocuvirus*, *Begomovirus*, and some virus species later classified into *Maldovirus*. The amino acid of the OpV1 Rep is revealed to form a clade with a cluster containing *Curtovirus, Topocuvirus, Turncurtovirus, Maldovirus,* and *Begomovirus,* similar to the findings from Fontenele *et al.* [49]. The constructed phylogeny using amino acids of the CP shows OpV1 CP to be closely related to the genus *Grablovirus* CP. This is contrary to the report of OpV1 CP forming a distinct clade [49].

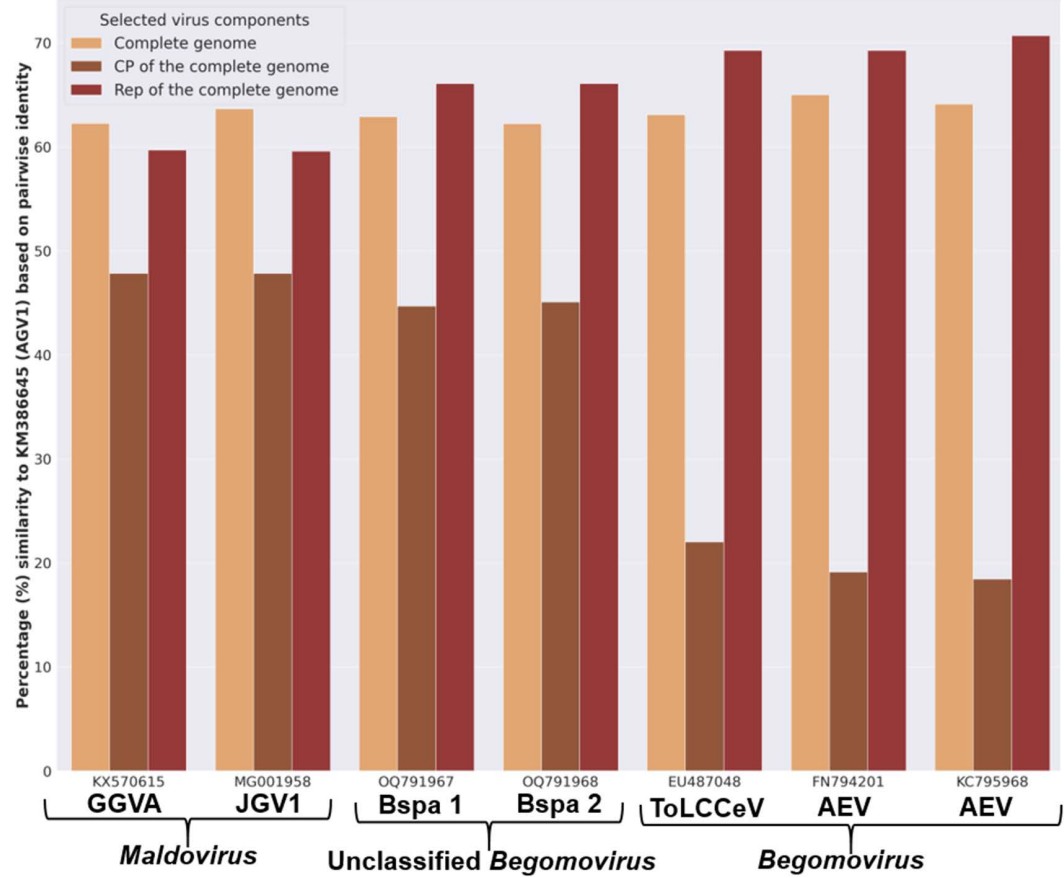

**Fig 11. Comparison of the pairwise identity of the newly proposed species of *Maldovirus* to *Begomovirus* and *Maldovirus*.**

*Topilevirus* genome and amino acid sequences of the Rep have close a resemblance to *Capulavirus,* while amino acid sequences of the CP are closely related to *Topocuvirus.* These findings on members within the genus *Topilevirus* are consistent with the previous studies [43,50]. The CP amino acids of the members within *Capulavirus* form distinct clusters that distantly form a clade with clusters containing *Begomovirus, Mulcrilevirus,* and *Citlodavirus*. This is similar to the isolated cluster of *Capulavirus* CP in the reported phylogeny by Roumagnac *et al.* [43] and Vaghi Medina *et al.* [50].

Genus *Grablovirus* is related to *Capulavirus, Topilevirus,* and *Mastrevirus,* consistent with the result of Vaghi Medina *et al.* [50]. The amino acids of the CP-constructed phylogeny reveal this virus to be closely related to the genus Opunvirus. However, few previous studies reported geminivirus phylogeny containing *Opunvirus* and *Grablovirus*. Roumagnac *et al.* [43] had these two genera included in their study, but the *Grablovirus* was revealed to be distinctly isolated. Interestingly, this study reveals the Rep forming a sister group to the remaining genera of the family *Geminiviridae.* This finding is consistent with the <45% pairwise identity of *Grablovirus* Rep with other geminivirus Rep [51].

The complete genomes of monopartite begomoviruses and DNA-A of bipartite are unveiled to be closely related to the maldoviruses. This is consistent with previous studies carried out by Roumagnac *et al.* [43] and Varsani *et al.* [51]. However, the amino acids of several begomoviruses CPs are closely related to CPs of citlodaviruses and mulcrileviruses. In contrast, the CP of a few viruses whose sequences are within *Begomovirus* in the GenBank, such as Parsley yellow leaf curl virus (PYLCV) and Sida golden yellow spot virus, are distant from the majority of the begomoviruses. PYLCV CP is

closely related to *Turncurtovirus, Becurtovirus,* and *Curtovirus,* similar to the reported findings by Hasanvand *et al.* [52]. Nichkerdar et al. demonstrated the virus to be transmitted by leafhoppers [53]. Previous studies reported the *Begomovirus* CP to be closely related to the *Citlodavirus* and *Mulcrilevirus* CP [43,51]. Lastly, the Rep amino acids of the begomoviruses are distributed across various genera such as *Maldovirus, Curtovirus, Turncurtovirus,* and *Topocuvirus,* supporting potential evolutionary relationships among these genera. However, only maldoviruses and topocuviruses were shown to be closely related in previous studies [43,51]. The differences in this study in comparison with previous studies could rely majorly on the representative sampling of the entire virus members.

The CP region has been demonstrated to confer vector specificity on geminivirus [54]. Additionally, the evolutionary relationships of geminivirus CPs and the cytochrome oxidase I (COI) gene of their insect vectors are closely aligned, providing strong evidence for reciprocal co-evolution between geminiviruses and their respective vectors [46]. Hence, evolutionary shifts in vector-mediated transmission of geminiviruses to new plant hosts appear to be more possible than the adaptation of known geminiviruses to novel insect vectors. This insight enhances our ability to predict potential vectors for geminivirus genera for which vector information is currently lacking. It is hypothesized that *Opunvirus*, *Welwivirus*, and *Topilevirus* may be transmitted by treehoppers, whereas *Citlodavirus* and *Eragrovirus* are likely transmitted by leafhoppers. The potential vector for *Maldovirus*, however, remains uncertain.

Comparative phylogenetic analyses based on the complete genome, CP, and Rep revealed that the CP-based phylogeny exhibited the highest degree of conservation across the family. In contrast, the Rep-based phylogeny indicated a high level of diversity, particularly among begomoviruses Reps distributed across multiple genera. Notably, recombination analysis identified the highest frequency of recombination events within the genomic region encoding the Rep.

Several previous studies had also reported interspecies recombination in *Begomovirus* [15,21,55]. For inter-genera recombination, *Opunvirus* was detected to contain 693 recombinant sequences, of which *Begomovirus* contributed the major parental sequences. However, Fontelele *et al*. reported only intra-species recombination for members of the genus *Opunvirus* in their recombination detection analysis [49]. *Topocuvirus* was also detected to contain intergeneric recombinant sequences of 1144 bases. The major parental sequence is the *Begomovirus,* while the minor parental sequence is the Myrica rubra citlodavirus 1 (MRV1). Padidam *et al*. [21] and Varsani *et al*. [55] reported that the tomato pseudo-curly top virus was involved in the recombination for the evolution of geminivirus. Members of the *Curtovirus* genus have recombinant sequences from *Becurtovirus* as the major parent and Juncus maritimus geminivirus 1 of *Maldovirus* as the minor parent. Several studies had previously detected recombination in the genus *Curtovirus* [21,55,56].

The extensive recombination detected across *Begomovirus*, *Curtovirus*, *Topocuvirus*, and *Opunvirus* highlights the evolutionary fluidity and plasticity within *Geminiviridae*. Specifically, the detection of intergeneric recombination events where *Begomovirus* contributed parental sequences to various genera, including *Opunvirus* and *Topocuvirus* genomes, reinforces its central role in geminivirus evolution [21]. These recombination events may facilitate cross-species transmission, adaptation to new hosts, and expansion into novel ecological niches [45].

Summarily, these results indicate that *Geminiviridae* evolution is driven by a combination of ancient divergence, recombination, and potential vector shifts. The phylogenetic inconsistencies between CPs and Reps further support the modular evolution of geminivirus genomes, where different genomic components may evolve independently due to differential selective pressures [19]. This modularity could also explain the potential of viruses such as those observed in *Begomovirus*, where CPs and Reps cluster with different genera, enabling ecological versatility and rapid evolutionary responses.

## Conclusions

This study provides a comprehensive, family-wide assessment of the genetic diversity, phylogeny, and global distribution of the *Geminiviridae*. Inter-genus and interspecies pairwise identity analyses support the overall robustness of existing taxonomic classifications while also identifying notable exceptions that warrant taxonomic revision. The global distribution map highlights the widespread presence of *Begomovirus* and *Mastrevirus*, consistent with their broad host ranges and

effective vector-mediated transmission. Phylogenetic analyses reveal the modular architecture of geminivirus genomes, with coat protein sequences exhibiting higher conservation—likely reflecting vector specificity—while replication-associated protein sequences display greater variability, driven largely by recombination. Recombination is confirmed as a major force shaping geminivirus evolution, particularly within *Begomovirus*, enabling rapid genetic diversification, host adaptation, and ecological expansion. These findings enhance our understanding of the evolutionary dynamics within *Geminiviridae* and underscore the importance of global surveillance, especially in regions with high viral diversity. The insights gained provide a critical foundation for improving the prediction of virus emergence, elucidating vector-virus relationships, and developing targeted disease management strategies essential for protecting global agricultural systems.

## Supporting information

**S1 Fig. Sampling approach based on pairwise identity similarity.** The bar chart depicted the retrieved and sampled nucleotide sequences across each genus.
(TIF)

**S2 Fig. Sampled coat and Rep proteins from each genus.**
(TIF)

**S1 Table. The details of the selected geminiviruses.** These geminiviruses were selected following the partition that was based on the pairwise identity.
(XLSX)

**S2 Table. *Geminiviridae* genera and respective countries data points for map construction.**
(XLSX)

**S3 Table. Interspecies and intraspecies recombination events in the sampled monopartite and *Begomovirus* DNA-A.** The methods with the lowest and highest P-values are superscripted with l and h respectively.
(XLSX)

**S4 Table. Interspecies recombination events in the sampled *Begomovirus* DNA-B.** The methods with the lowest and highest P-values are superscripted with l and h respectively.
(XLSX)

**S5 Table. Inter-genera recombination events in the sampled monopartite and *Begomovirus* DNA-A.** The methods with the lowest and highest P-values are superscripted with l and h respectively.
(XLSX)

## Author contributions

**Conceptualization:** Moshood Olamide Lateef, Neil Bretana.

**Data curation:** Moshood Olamide Lateef.

**Formal analysis:** Moshood Olamide Lateef.

**Methodology:** Moshood Olamide Lateef, Dauda Nathaniel, Bolaji Osundahunsi, Neil Bretana.

**Supervision:** Neil Bretana.

**Visualization:** Moshood Olamide Lateef, Dauda Nathaniel, Bolaji Osundahunsi.

**Writing – original draft:** Moshood Olamide Lateef, Dauda Nathaniel.

**Writing – review & editing:** Moshood Olamide Lateef, Dauda Nathaniel, Bolaji Osundahunsi, Neil Bretana.

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
