## [Decision Letter · Decision Letter 0]

10 Jun 2025

Dear Dr. Bretana,

Thank you for submitting your manuscript to PLOS ONE. After careful consideration, we feel that it has merit but does not fully meet PLOS ONE’s publication criteria as it currently stands. Therefore, we invite you to submit a revised version of the manuscript that addresses the points raised during the review process.

Thank you for submitting your manuscript to *PLOS ONE* . Your study presents a large-scale comparative analysis of the Geminiviridae family, utilizing extensive genomic data and comprehensive bioinformatics approaches. This topic is timely and highly relevant to the fields of plant virology and molecular evolution, particularly in understanding viral diversity, recombination, and taxonomy.We have received detailed evaluations from two expert reviewers who acknowledge the potential significance of your work and appreciate the valuable dataset and analyses you have provided. However, both reviewers have identified several substantive concerns that must be addressed before the manuscript can be considered for publication.Summary of Key Concerns:The abstract requires revision to better emphasize the biological significance of your findings, rather than focusing heavily on specific software tools.There is inconsistent use of terminology and viral taxonomy, including species names and gene/protein nomenclature, which should conform strictly to ICTV standards.The methods section lacks sufficient detail, especially regarding outgroup selection, sequence alignment strategies, and data analysis procedures, which affects reproducibility and transparency.While comprehensive, the results section needs clearer organization into thematic subsections, improved figure quality, and more informative legends to enhance reader comprehension.The discussion should be substantially expanded, offering deeper biological interpretation, evolutionary insights, and stronger integration with existing literature to fully convey the importance of your findings.The manuscript requires a thorough language and grammar revision to improve readability and clarity.Lastly, a detailed, point-by-point response to all reviewer comments will be necessary to demonstrate how the manuscript has been improved.We invite you to revise your manuscript addressing these points carefully. Key areas for revision include:Rewriting the abstract and introduction with a clear focus on biological insights.Ensuring consistent and accurate use of viral taxonomy and nomenclature throughout the manuscript.Expanding the methodological descriptions to ensure full reproducibility.Reorganizing the results section into clear subsections and enhancing figure quality.Strengthening the discussion with biological interpretation, evolutionary implications, and comparisons to prior research.Conducting thorough language editing to improve readability.Preparing a comprehensive, point-by-point response to reviewer comments.We believe that after these major revisions, your manuscript has strong potential to make a valuable contribution to the fields of plant virology and viral taxonomy. We look forward to receiving your revised manuscript for further consideration.Sincerely,Islam HamimAcademic Editor*PLOS ONE*

We look forward to receiving your revised manuscript.

Kind regards,

Islam Hamim, PhD

Academic Editor

PLOS ONE

Journal Requirements:

2. We note that your Data Availability Statement is currently as follows: All relevant data are within the manuscript and in Supporting Information files.

4. We note that Figure 3 in your submission contain map/satellite images which may be copyrighted. All PLOS content is published under the Creative Commons Attribution License (CC BY 4.0), which means that the manuscript, images, and Supporting Information files will be freely available online, and any third party is permitted to access, download, copy, distribute, and use these materials in any way, even commercially, with proper attribution. For these reasons, we cannot publish previously copyrighted maps or satellite images created using proprietary data, such as Google software (Google Maps, Street View, and Earth). For more information, see our copyright guidelines: http://journals.plos.org/plosone/s/licenses-and-copyright.

 1. You may seek permission from the original copyright holder of Figure 3 to publish the content specifically under the CC BY 4.0 license. 

Additional Editor Comments:

Dear Authors,

Thank you for submitting your manuscript to PLOS ONE. Your study presents a large-scale comparative analysis of the Geminiviridae family, utilizing extensive genomic data and comprehensive bioinformatics approaches. This topic is timely and highly relevant to the fields of plant virology and molecular evolution, particularly in understanding viral diversity, recombination, and taxonomy.

We have received detailed evaluations from two expert reviewers who acknowledge the potential significance of your work and appreciate the valuable dataset and analyses you have provided. However, both reviewers have identified several substantive concerns that must be addressed before the manuscript can be considered for publication.

Summary of Key Concerns:

• The abstract requires revision to better emphasize the biological significance of your findings, rather than focusing heavily on specific software tools.

• There is inconsistent use of terminology and viral taxonomy, including species names and gene/protein nomenclature, which should conform strictly to ICTV standards.

• The methods section lacks sufficient detail, especially regarding outgroup selection, sequence alignment strategies, and data analysis procedures, which affects reproducibility and transparency.

• While comprehensive, the results section needs clearer organization into thematic subsections, improved figure quality, and more informative legends to enhance reader comprehension.

• The discussion should be substantially expanded, offering deeper biological interpretation, evolutionary insights, and stronger integration with existing literature to fully convey the importance of your findings.

• The manuscript requires a thorough language and grammar revision to improve readability and clarity.

• Lastly, a detailed, point-by-point response to all reviewer comments will be necessary to demonstrate how the manuscript has been improved.

We invite you to revise your manuscript addressing these points carefully. Key areas for revision include:

• Rewriting the abstract and introduction with a clear focus on biological insights.

• Ensuring consistent and accurate use of viral taxonomy and nomenclature throughout the manuscript.

• Expanding the methodological descriptions to ensure full reproducibility.

• Reorganizing the results section into clear subsections and enhancing figure quality.

• Strengthening the discussion with biological interpretation, evolutionary implications, and comparisons to prior research.

• Conducting thorough language editing to improve readability.

• Preparing a comprehensive, point-by-point response to reviewer comments.

We believe that after these major revisions, your manuscript has strong potential to make a valuable contribution to the fields of plant virology and viral taxonomy. We look forward to receiving your revised manuscript for further consideration.

Sincerely,

Islam Hamim

Academic Editor

PLOS ONE

Reviewers' comments:

Reviewer's Responses to Questions

**Comments to the Author**

1. Is the manuscript technically sound, and do the data support the conclusions?

Reviewer #1: Yes

Reviewer #2: Yes

2. Has the statistical analysis been performed appropriately and rigorously?

Reviewer #1: Yes

Reviewer #2: Yes

3. Have the authors made all data underlying the findings in their manuscript fully available?

Reviewer #1: Yes

Reviewer #2: Yes

4. Is the manuscript presented in an intelligible fashion and written in standard English?

Reviewer #1: Yes

Reviewer #2: Yes

Reviewer #1: This article presents an interesting contribution regarding the importance of regularly updating public databases such as GenBank, or at least exercising caution when using them in phylogenetic studies focused on a single genus. However, the manuscript still requires substantial improvements.

Firstly, I was somewhat disappointed by the use of the term "data mining" in the title, especially considering the limited explanation of the methodology. Nowadays, this term is increasingly used across disciplines, and I recommend being cautious with its application. I am not fully convinced that its usage here is entirely appropriate given the methods described. If the authors are confident in their choice, they should clearly justify it in the manuscript and explain its relevance and implications.

Secondly, the abstract needs to be entirely rewritten. Several elements mentioned there are not discussed or even mentioned in the main text, while key findings from the conclusion should instead be integrated into the discussion. As it stands, the discussion is primarily a comparison with previous studies and lacks depth. For instance, the authors should explore what could explain the differences observed.

Moreover, there is some confusion and inconsistency in the use of viral nomenclature throughout the text. I strongly recommend checking and adhering to ICTV rules. For example, genus and family names must be italicized and capitalized (e.g., Topocuvirus genus, Geminiviridae family), whereas common names should be written in lowercase and not italicized (e.g., the geminiviruses). Furthermore, French bean severe leaf curl virus is the common name of the species Capulavirus phaseoli and should appear in regular font. This rule applies to all species names; however, if a word in the name is Latin (e.g., Corchorus yellow vein mosaic virus), that word (Corchorus) should be capitalized. Similar attention should be paid to gene or protein names: rep and cp (gene names) versus Rep and CP (protein names), depending on the context.

Here are my recommendations throughout the manuscript:

Title: Is “data mining” really the most appropriate term here? Please discuss its usage and consider whether it accurately reflects the methods employed.

Abstract: Avoid including specific methodological tools such as SDT, MAFFT, IQ-TREE, and RDP5. Instead, provide a general overview of the methodology (e.g., mention a data mining approach, if appropriate) and focus more on the key findings. You may also want to add: "To the best of our knowledge, this is the first geographical map constructed using all genera from the Geminiviridae family."

Line 41 : why are they "recommended" ??

Line 60: Include more recent references to reflect the current state of knowledge.

Line 64: Cite relevant publications for each mechanism mentioned. Note that reassortment applies specifically to segmented viruses, which is not the case for members of the Geminiviridae family—please correct or clarify this point.

Line 76: Use either Geminiviridae family or geminiviruses, depending on whether you're referring to the taxonomic family or the viruses in general.

Line 95: What is the purpose of mentioning PhyloPart here? Is it relevant to the current study?

Lines 97–98: A period or comma appears to be missing before “Bandoo et al. adopted 80% pairwise identity in selecting members of begomoviruses [27]”. Also, this sentence would be better placed with the following paragraph that discusses begomoviruses.

Line 108: Rephrase as follows: “Similarly, DNA B sequences from bipartite species of Begomovirus — the only genus within the Geminiviridae family known to include bipartite viruses — were also downloaded.”

Line 114: Please specify which outgroup was used for the phylogenetic analyses.

Line 116: Why was the --adjustdirection option of MAFFT not used? This would ensure proper sequence orientation.

Lines 125–126: Rephrase as: “For all genera, as well as for the DNA-A of bipartite begomoviruses, aligned nucleotide sequences of complete genomes and amino acid sequences of Rep and CP were used to reconstruct phylogenetic trees using IQ-TREE...”

Line 130: Is the outgroup mentioned here the same as the one used in line 114? If so, make this clear.

Line 152: How were 28,185 data points generated from 17,718 sequences (mentioned in line 89)? Please clarify the calculation or criteria used.

Line 174: Use “Rep and CP” for consistency.

Line 177: At the first mention of these proteins, spell out their full names followed by the abbreviations in parentheses, e.g., replication-associated protein (Rep) and coat protein (CP).

Line 201: Specify whether the nucleotide sequences used were full genomes or just gene fragments. Also indicate whether BLASTn, BLASTx, or both were used, and which reference database was used.

Line 205: Why was ClustalW used for alignment when MAFFT is available within SDT? Using MAFFT would provide consistency across the methods section.

Lines 231–232: This information should also be included in the abstract, as it represents a key result.

Discussion: Before comparing similarities across genera (line 235), discuss possible explanations for the absence of certain genera in specific regions of the world.

In connection with the final sentences of the abstract, you should also discuss the potential role of viral vectors in shaping CP similarities. A useful reference could be: https://doi.org/10.1038/s41579-019-0232-3

Line 325: Avoid citing references in the conclusion. Instead, move the cited discussion to the appropriate section earlier in the manuscript.

Conclusion: Much of the content currently in the conclusion (at least from lines 324 to 330) should be relocated to the discussion. The conclusion should instead summarize the key findings and emphasize the value of using large-scale, database-driven (“data mining”) approaches in viral diversity studies.

Reviewer #2: This manuscript focuses on a comprehensive investigation of genetic diversity, recombination events, and taxonomic misclassifications within the Geminiviridae family. The authors utilize data mining and various bioinformatics tools to analyze 17,718 complete genome sequences retrieved from GenBank, covering all genera within this viral family. Through the use of SDT, MAFFT, IQ-tree, and RDP5, the study performs sequence similarity partitioning, phylogenetic tree construction, and recombination detection. While the manuscript addresses a relevant topic and involves the analysis of a large dataset using diverse tools, several major issues must be addressed to improve the scientific rigor and clarity of the work.

Below are some suggestions intended to improve the clarity and phrasing of the manuscript:

Line 46: "It exists in monopartite or bipartite..." should be "It exists as monopartite or bipartite..."

Line 47: "They cause huge losses to several plant hosts..." change to "These viruses cause huge losses to various plant hosts..."

Line 50: “Asian nations like Pakistan and India" change to "Asian countries like Pakistan and India"

Line 51: “The cotton industry in Pakistan faces losses of $5 billion in Pakistan” revise to:

"The cotton industry in Pakistan faces losses of $5 billion annually."

Line 55: "Making it the family with the largest plant viruses" revise to "Making it the largest family of plant viruses."

Line 59: "Which is made possible because of the sequencing revolution..." revise to "Due to advances in sequencing technology..."

Line 61: "With a single insect vector capable of transmitting to multiple species or genera to the susceptible host" revise to "With a single vector capable of transmitting viruses to multiple species or genera of susceptible hosts."

Line 68: "Recombination has been the driving force behind the evolution and generation of new viruses within the geminiviruses..." revise to "Recombination has been the driving force behind the evolution and emergence of new viruses within the Geminiviridae family..."

Line 70: "…towards the emergence of new viruses within the Geminiviridae" revise to "leading to the emergence of new viruses within Geminiviridae"

Line 72: "Hence, there is a need to consider all 15 genera of Geminiviridae..." revise to "Therefore, all 15 genera of Geminiviridae need to be considered..."

Line 73: "to reveal a holistic approach to how the entire family possibly recombines" revise to "to provide a comprehensive view of recombination across the entire family"

Line 75: "as a representative means of revealing the true diversity of this known diverse virus family" revise to "to represent the full sequence diversity of this highly diverse virus family"

Line 76: "The distribution of the entire genera within Geminiviridae is reported to be global" revise to "The distribution of all Geminiviridae genera is reported to be global"

Line 78: "A geographical map revealing the distribution of the entire genera across all the continents is still missing" revise to "A comprehensive map showing the global distribution of all genera is still lacking"

Line 80: "Understanding the diversity and evolution of the new and previously reported species..." revise to "Understanding the diversity and evolution of both new and previously reported species..."

Line 81: "to ensure surveillance action towards taming the prospective diseases that endanger food security" revise to "to support surveillance efforts against emerging diseases threatening food security"

Line 82: "status of diversity and recombination within and between all the 15 genera" revise to "patterns of diversity and recombination within and among all 15 genera"

Please correct "GeneBank" to "GenBank" in lines 88 and 334.

Line 87: Change "the entire virus sequence of every monopartite and bipartite" to "all available monopartite and bipartite viruses" for clarity and to avoid redundancy and changed the verb from singular "was" to plural "were" to agree with the plural subject.

Line 88: Replace "This amounted to" with "This included" for clearer and more natural phrasing.

Line 89: Clarify "of monopartite and DNA-A of bipartite" to "of monopartite viruses and DNA-A components of bipartite viruses" for precision and readability.

Line 90: Change "the chart depicted" to "A chart showing" to avoid unnecessary passive voice and improve sentence flow.

Line 91: Revise "included in the supplementary (S Figure. 1,2)" to "is included in the supplementary materials (S1 and S2 Figs)".

Line 103: Change "with large available sequences" to "which had a large number of available sequences"

Line 104: Change "their respective sequences were grouped into batches to perform parallel computation" to "the sequences were grouped into batches to enable parallel computation"

Line 106: Change "served as the representative for the particular genus" to "used as representatives for their respective genera".

Line 106: Change "respective amino acid sequences" to "corresponding amino acid sequences"

Please capitalize "rep" to "Rep" consistently throughout the manuscript.

Line 108: Revise "Similarly, the DNA B of the bipartite Geminiviridae (begomovirus)" to "Similarly, DNA-B sequences of bipartite Geminiviridae (i.e., Begomovirus)"

Line 109: Reworded "can be found in the supplementary (S Table 1)" to "are provided in the supplementary materials (S Table 1)"

Line 111: Change "imported to a single FASTA file" to "imported into individual FASTA files"

Line 114: Change "with and without outgroup sequence" to "with and without outgroup sequences"

Line 116: Change "the wrong orientation" to "incorrect orientation"

Line 120: "collapsed to their respective" Change to "aggregated by their respective"

Line 121: "to avoid graphics cluttering" Replace with "to avoid graphic clutter"

Line 122: "based on Python programming" Change to "in Python"

Line 124: The evolutionary relationship among the entire genera" Replace with "The evolutionary relationships among all genera"

Line 127: "automatic selection of the substitution model that is best fit" Replace with "best-fitting substitution model was automatically selected"

Line 134: "Alignment of the multiple sequences" Replace with "The multiple sequence alignments"

Line 134: "was exported into" revise to "were imported into"

Line 138: Change "default setting parameters" to "default parameter settings"

Line 139: "To confer more stringency on detecting false positive recombination" Replace with "To increase the stringency and reduce false positives"

Please revise all occurrences of "pairwise identity similarity" throughout the manuscript. The correct and standard term is simply "pairwise identity". Adding "similarity" is redundant and grammatically incorrect in this context.

Please ensure that all virus genus names (e.g., Begomovirus, Mastrevirus, Curtovirus, Citlodavirus, Grablovirus, Topocuvirus, Turncurtovirus, etc.) are consistently formatted throughout the manuscript. As per scientific conventions, genus names must be italicized and begin with a capital letter.

In addition, according to virus nomenclature, the name of the virus (e.g., French bean severe leaf curl virus) should not be italicized.

Line 154: Change "However, begomovirus and mastrevirus remain the only genera in this family that were isolated from all the continents" to "However, only the genera Begomovirus and Mastrevirus have been isolated from every continent."

Line 165: Change "Begomovirus, mastrevirus, and maldovirus were the only three reported genera isolated from Oceania" to "Only three genera—Begomovirus, Mastrevirus, and Maldovirus—have been reported from Oceania"

Lines 171-173, Change to "The total number of sampled sequences used to construct the phylogenetic trees was 284 for the complete genome, 280 for the coat protein, and 282 for the Rep protein. After multiple sequence alignment, the orientation of all sequences was checked and confirmed"

Line 193: Change "Identifying and Analyzing the Misclassified Virus Accession" to "Identifying and Analyzing a Misclassified Viral Accession"

Line 195: Add commas after "KC699544" and "Capulavirus"

Line 201: Change "blasted to" to "blasted against"

Line 203: Change "result" to "results" when referring to BLAST outcomes.

Line 208: Change "Explicitly, to visualized" to "To explicitly visualize."

Line 210: "…used to plot a multiple bar chart for the nucleotide, cp, and rep as shown in Fig 10." Change to "…used to plot a multiple bar chart for the complete genome, CP, and Rep regions, as shown in Fig. 10."

Line 209: Revise "percentage similarity score was extracted" to "scores were extracted".

Line 224: Change "Detected interspecies recombinations are of higher numbers in the begomovirus" to "Interspecies recombinations were detected in higher numbers in Begomovirus."

Line 226: Change "The tables that detailed" to "Tables detailing."

Line 230: Change "This adopted sampling approach in this study considers..." to "The sampling approach adopted in this study considers..."

Line 244: Change "The complete genome and coat protein of the genus Eragrovirus are more related..." to "The complete genome and coat protein of the genus Eragrovirus were more closely related…"

Line 251: "The genome of the genus Welwivirus forms a clade with the topileviruses..." Change to "The genome of the genus Welwivirus formed a clade with the topileviruses..."

Line 271: Revise "is more related to" to "is more closely related to"

Line 252: Change "Contrarily, welwivirus CP forms a clade with cluster of topileviruses..." to "In contrast, the Welwivirus CP forms a clade with a cluster of topileviruses..."

Line 271: "The genome... are more related" Should be "The genome... is more closely related"

Line 304: Please Correct "Fiallo-Olivé are Navas-Castillo" to "Fiallo-Olivé and Navas-Castillo"

Line 304: "Paddidam" should be corrected to "Padidam"

It is worth mentioning that in Geminiviruses (Geminiviridae), viral replication relies on the host DNA polymerase, while the viral Replication-associated protein (Rep) plays a key role in initiating and facilitating the replication process (Not Replicase). Therefore, correct the legend of Figure 10 i.e., the text terminology of the color labels within the plot itself. The term "Replicase of the complete genome" is incorrect and should be replaced with "Replication-associated protein (Rep)", which is the accurate term used in Geminiviruses.

All figures appear to be of low resolution and lack sufficient clarity. While this may ultimately be addressed by the journal's editorial office, authors are encouraged to submit high-quality versions for better readability.

Although recombinant events are identified, their biological or phylogenetic implications are not well contextualized. Moreover, the potential influence of recombination on tree topology or taxonomic misassignment is not fully explored.

Inconsistencies were observed in the reference list, with some entries including the month of publication while others did not. Please remove all months from the references to comply with PLOS ONE guidelines, ensure journal names are properly abbreviated, and maintain consistent formatting throughout.

**Do you want your identity to be public for this peer review?** For information about this choice, including consent withdrawal, please see our Privacy Policy

Reviewer #1: No

Reviewer #2: No

---

## [Author Response · Author response to Decision Letter 1]

18 Jul 2025

Reviewer 1

Comment 1: Firstly, I was somewhat disappointed by the use of the term "data mining" in the title, especially considering the limited explanation of the methodology. Nowadays, this term is increasingly used across disciplines, and I recommend being cautious with its application. I am not fully convinced that its usage here is entirely appropriate given the methods described. If the authors are confident in their choice, they should clearly justify it in the manuscript and explain its relevance and implications. Secondly, the abstract needs to be entirely rewritten. Several elements mentioned there are not discussed or even mentioned in the main text, while key findings from the conclusion should instead be integrated into the discussion. As it stands, the discussion is primarily a comparison with previous studies and lacks depth. For instance, the authors should explore what could explain the differences observed. Moreover, there is some confusion and inconsistency in the use of viral nomenclature throughout the text. I strongly recommend checking and adhering to ICTV rules. For example, genus and family names must be italicized and capitalized (e.g., Topocuvirus genus, Geminiviridae family), whereas common names should be written in lowercase and not italicized (e.g., the geminiviruses). Furthermore, French bean severe leaf curl virus is the common name of the species Capulavirus phaseoli and should appear in regular font. This rule applies to all species names; however, if a word in the name is Latin (e.g., Corchorus yellow vein mosaic virus), that word (Corchorus) should be capitalized. Similar attention should be paid to gene or protein names: rep and cp (gene names) versus Rep and CP (protein names), depending on the context.

Response 1: We appreciate all of these comments. However, we addressed the ICTV rules regarding the naming of species, genus, and family only here because the remaining comments were specifically listed and will be addressed accordingly. The rules of naming the viruses have been applied throughout this revised manuscript.

Change Made: This change occurs throughout the content of this revised manuscript.

Comment 2: Title: Is “data mining” really the most appropriate term here? Please discuss its usage and consider whether it accurately reflects the methods employed.

Response 2: This suggestion is highly positive and appreciated. Inclusion of data mining was used due to the utilization of data preprocessing mainly on the metadata, a step in the data mining workflow. However, we have decided to remove the term provided that we did not use the entire data mining workflow in this study.

Change Made: The term ‘data mining’ was removed (line 1).

Comment 3: Abstract: Avoid including specific methodological tools such as SDT, MAFFT, IQ-TREE, and RDP5. Instead, provide a general overview of the methodology (e.g., mention a data mining approach, if appropriate) and focus more on the key findings. You may also want to add: "To the best of our knowledge, this is the first geographical map constructed using all genera from the Geminiviridae family."

Response 3: This wonderful observation is appreciated for exploring the attractiveness of this work. We agreed and rewrote the abstract with the inclusion of more key findings.

Change Made: Described in lines 21-46.

Comment 4: Line 41: why are they "recommended”??

Response 4: They were recommended because of the relationship of geminivirus coat protein with vector specificity. We understood the reason for this laudable comment, and we have included the details of this vector specificity in this revised manuscript.

Change Made: Described in lines 340-348.

Comment 5: Line 60: Include more recent references to reflect the current state of knowledge.

Response 5: This is well appreciated; more references have been included.

Change Made: Described in line 63.

Comment 6: Line 64: Cite relevant publications for each mechanism mentioned. Note that reassortment applies specifically to segmented viruses, which is not the case for members of the Geminiviridae family—please correct or clarify this point.

Response 6: Although the inclusion of the reassortment in the references was to capture the evolution mechanism for all viruses (segmented and non-segmented viruses), however, to avoid misinformation in this present study, we agreed and decided to remove the reassortment and update it with appropriate publication.

Change Made: Described in line 67.

Comment 7: Line 76: Use either Geminiviridae family or geminiviruses, depending on whether you're referring to the taxonomic family or the viruses in general.

Response 7: The taxonomic family for all the genera was upheld, and the sentence was rewritten.

Change Made: Described in line 79.

Comment 8: Line 95: What is the purpose of mentioning PhyloPart here? Is it relevant to the current study?

Response 8: This sentence basically suggests some other tools being used aside from SDT for pairwise identity calculation. However, for the tool not being used for this study, we decided to remove the sentence.

Change Made: The sentence was removed.

Comment 9: Lines 97–98: A period or comma appears to be missing before “Bandoo et al. adopted 80% pairwise identity in selecting members of begomoviruses [27]”. Also, this sentence would be better placed with the following paragraph that discusses begomoviruses.

Response 9: This observation is well appreciated, and we have included the missing period,and the sentence is placed into the paragraph that discusses begomoviruses.

Change Made: Described in lines 98 and 105-106.

Comment 10: Line 108: Rephrase as follows: “Similarly, DNA B sequences from bipartite species of Begomovirus — the only genus within the Geminiviridae family known to include bipartite viruses — were also downloaded.”

Response 10: The sentence that required this comment was removed because DNA-A and DNA-B were downloaded together and separated with SDT. This is detailed in the shared protocol. We have to remove it to avoid confusing readers.

Change Made: Sentence removed.

Comment 11: Line 114: Please specify which outgroup was used for the phylogenetic analyses.

Response 11: Chicken anaemia virus with accession number M55918.1 was the used outgroup, previously adopted in previous studies [3, 30].

Change Made: Described in lines 116-118.

Comment 12: Why was the --adjustdirection option of MAFFT not used? This would ensure proper sequence orientation.

Response 12: We sincerely appreciate the reviewer’s This suggestion; however, we opted not to use the proposed option for the following reasons:

1. Sensitivity to Divergent Sequences: As clearly indicated on the software’s official documentation, the method performs reliably except when sequences are highly divergent: “It works well unless the sequences are highly diverged.” Geminiviruses, particularly at the family level, are known to exhibit high genetic diversity.

2. Intended Purpose of the option: The option is primarily designed to reverse sequences to the same direction (5′ to 3′). It is not optimized for reorienting circular genomes

Based on these considerations, we instead employed manual reorientation of sequences suspected to be misaligned. Following the reviewer’s valuable comment, we revisited the alignment, specifically focusing on nucleotide sequences that appeared phylogenetically misplaced. We reoriented a few such sequences, repeated the multiple sequence alignment, and reconstructed the phylogenetic tree.

While only minor changes were observed in the overall phylogeny compared to the previously submitted nucleotide-based tree, this reanalysis led to the identification of a potential new species within the genus Maldovirus. Notably, these new species (Begomovirus spathoglottis 1 and 2) were also present in the original phylogeny, but their potential novelty became more evident upon closer scrutiny during this reanalysis.

We have now included pairwise identity calculations for these new species in the revised manuscript. Additionally, we re-evaluated the pairwise identity for the misclassified Capulavirus due to one of virus accession that fell outside the genus in the nucleotide-based phylogeny.

Change Made: The result of these new species was included in the lines 231-248. Two figures were also updated (Figs 4 and 7) from the repeated analyses.

Comment 13: Lines 125–126: Rephrase as: “For all genera, as well as for the DNA-A of bipartite begomoviruses, aligned nucleotide sequences of complete genomes and amino acid sequences of Rep and CP were used to reconstruct phylogenetic trees using IQ-TREE...”

Response 13: This suggestion is well appreciated and has been implemented accordingly.

Change Made: Described in lines 130-132.

Comment 14: Line 130: Is the outgroup mentioned here the same as the one used in line 114? If so, make this clear.

Response 14: Yes, it is the same outgroup and has been made clear in lines 116-118. Additionally, it was deleted here to avoid repetition, and the sentence was rephrased.

Change Made: Removed to avoid repetition.

Comment 15: Line 152: How were 28,185 data points generated from 17,718 sequences (mentioned in line 89)? Please clarify the calculation or criteria used.

Response 15: 17,718 sequences are complete genomes only, while 28,185 data points are for both complete and partial genomes. This has been updated in this revised manuscript.

Change Made: Described in lines 159-160.

Comment 16: Line 174: Use “Rep and CP” for consistency

Response 16: This suggestion is well appreciated and has been implemented accordingly.

Change Made: This change occurs throughout the content of this revised manuscript.

Comment 17: Line 177: At the first mention of these proteins, spell out their full names followed by the abbreviations in parentheses, e.g., replication-associated protein (Rep) and coat protein (CP).

Response 17: This suggestion is well appreciated and has been implemented accordingly.

Change Made: Described in lines 109-110.

Comment 18: Line 201: Specify whether the nucleotide sequences used were full genomes or just gene fragments. Also indicate whether BLASTn, BLASTx, or both were used, and which reference database was used.

Response 18: Full genomes were used with BLASTN and BLASTP. This revised manuscript has been updated with this information.

Change Made: Described in line 207.

Comment 19: Line 205: Why was ClustalW used for alignment when MAFFT is available within SDT? Using MAFFT would provide consistency across the methods section.

Response 19: MAFFT was initially used but not running successfully with the selected few protein (CP and Rep) sequences (it gave run-time error ‘53’: file not found). To avoid the bias of running nucleotides with MAFFT and other algorithms for protein sequences, we opt to use ClustalW for all. This applies to the output of figure 11, where the default parameter (MUSCLE) was used, as stated in the manuscript.

Comment 20: Lines 231–232: This information should also be included in the abstract, as it represents a key result.

Response 20: This suggestion is well appreciated and has been included in the abstract.

Change Made: Described in lines 30-31.

Comment 21: Discussion: Before comparing similarities across genera (line 235), discuss possible explanations for the absence of certain genera in specific regions of the world.

Response 21: The possible explanation for the absence or presence of certain genera in specific regions of the world was discussed as suggested.

Change Made: Described in lines 265-286.

Comment 22: In connection with the final sentences of the abstract, you should also discuss the potential role of viral vectors in shaping CP similarities. A useful reference could be: https://doi.org/10.1038/s41579-019-0232-3

Response 22: This publication is well appreciated and related to the relationship of CP to vector specificity. Discussion on the potential role of viral vectors in shaping CP similarities has been updated in this revised manuscript.

Change Made: Described in lines 340-348.

Comment 23: Line 325: Avoid citing references in the conclusion. Instead, move the cited discussion to the appropriate section earlier in the manuscript.

Response 23: This is well appreciated and was moved to the discussion section.

Change Made: Described in line 340.

Comment 24: Conclusion: Much of the content currently in the conclusion (at least from lines 324 to 330) should be relocated to the discussion. The conclusion should instead summarize the key findings and emphasize the value of using large-scale, database-driven (“data mining”) approaches in viral diversity studies.

Response 24: As earlier suggested, these lines were relocated to the discussion. The conclusion was rewritten.

Change Made: The previous conclusion was deleted, and a new one was rewritten.

Reviewer 2

Comment 1: Line 46: "It exists in monopartite or bipartite..." should be "It exists as monopartite or bipartite..."

Response 1: This suggestion is well appreciated and updated accordingly.

Change Made: Described in line 49

Comment 2: Line 47: "They cause huge losses to several plant hosts..." change to "These viruses cause huge losses to various plant hosts..."

Response 2: This suggestion is well appreciated and updated accordingly.

Change Made: Described in lines 50-51.

Comment 3: Line 50: “Asian nations like Pakistan and India" change to "Asian countries like Pakistan and India"

Response 3: This suggestion is well appreciated and updated accordingly.

Change Made: Described in line 53.

Comment 4: Line 51: “The cotton industry in Pakistan faces losses of $5 billion in Pakistan” revise to: "The cotton industry in Pakistan faces losses of $5 billion annually."

Response 4: This suggestion is well appreciated and updated accordingly.

Change Made: Described in line 54.

Comment 5: Line 55: "Making it the family with the largest plant viruses" revise to "Making it the largest family of plant viruses."

Response 5: This suggestion is well appreciated and updated accordingly.

Change Made: Described in line 58.

Comment 6: Line 59: "Which is made possible because of the sequencing revolution..." revise to "Due to advances in sequencing technology..."

Response 6: This suggestion is well appreciated and updated accordingly.

Change Made: Described in line 62.

Comment 7: Line 61: "With a single insect vector capable of transmitting to multiple species or genera to the susceptible host" revise to "With a single vector capable of transmitting viruses to multiple species or genera of susceptible hosts."

Response 7: This suggestion is well appreciated and updated accordingly.

Change Made: Described in lines 64-65.

Comment 8: Line 68: "Recombination has been the driving force behind the evolution and generation of new viruses within the geminiviruses..." revise to "Recombination has been the driving force behind the evolution and emergence of new viruses within the Geminiviridae family..."

Response 8: This suggestion is well appreciated and updated accordingly.

Change Made: Described in lines 71-72.

Comment 9: Line 70: "…towards the emergence of new viruses within the Geminiviridae" revise to "leading to the emergence of new viruses within Geminiviridae"

Response 9: This suggestion is well appreciated and updated accordingly.

Change Made: Described in lines 73-74.

Comment 10: Line 72: "Hence, there is a need to consider all 15 genera of Geminiviridae..." revise to "Therefore, all 15 genera of Geminiviridae need to be considered..."

Response 10: This suggestion is well appreciated and updated accordingly.

Change Made: Described in line 75.

Comment 11: Line 73: "to reveal a holistic approach to how the entire family possibly recombines" revise to "to provide a comprehensive view of recombination across the entire family"

Response 11: This suggestion is well appreciated and updated accordingly.

Change Made: Described in line 76.

Comment 12: Line 75: "as a representative means of revealing the true diversity of this known diverse virus family" revise to "to represent the full sequence diversity of this highly diverse virus family"

Response 12: This suggestion is well appreciated and updated accordingly.

Change Made: Described in line 78.

Comment 13: Line 76: "The distribution of

---

## [Decision Letter · Decision Letter 1]

22 Sep 2025

Dear Dr. Bretana,

Thank you for submitting your manuscript to PLOS ONE. After careful consideration, we feel that it has merit but does not fully meet PLOS ONE’s publication criteria as it currently stands. Therefore, we invite you to submit a revised version of the manuscript that addresses the points raised during the review process.

We look forward to receiving your revised manuscript.

Kind regards,

Islam Hamim, PhD

Academic Editor

PLOS ONE

Journal Requirements:

Reviewers' comments:

Reviewer's Responses to Questions

**Comments to the Author**

Reviewer #1: All comments have been addressed

Reviewer #2: All comments have been addressed

2. Is the manuscript technically sound, and do the data support the conclusions?

Reviewer #1: Yes

Reviewer #2: Yes

3. Has the statistical analysis been performed appropriately and rigorously?

Reviewer #1: Yes

Reviewer #2: Yes

4. Have the authors made all data underlying the findings in their manuscript fully available?

Reviewer #1: Yes

Reviewer #2: Yes

5. Is the manuscript presented in an intelligible fashion and written in standard English?

Reviewer #1: Yes

Reviewer #2: Yes

Reviewer #1: The authors have convincingly addressed the points I raised.

I only have a few minor remaining comments:

L.52-56: Please cite the viral species responsible.

L.107: The link https://doi.org/10.17504/protocols.io.j8nlkybjdg5r/v1 does not work.

L.169-170: I suggest changing the sentence to:

"Additionally, the African continent had the highest number of countries where geminivirus infections are reported."

Instead of:

"Additionally, the African continent had the highest number of countries affected with geminivirus infections."

This avoids potential bias linked to differences in sampling effort, as more sampled countries are more likely to show higher numbers of infections or viral diversity.

Reviewer #2: The authors have addressed the reviewers’ comments thoroughly and provided convincing responses, leading to substantial improvement of the manuscript. In addition to these improvements, clarification is needed concerning the proposed virus names and their taxonomic status. The authors may propose names for new virus isolates, but it is the ICTV’s role to determine whether a new species should be formally recognized. Begomovirus spathoglottis 1 and 2 are introduced only in the Abstract and Results, with no prior mention in the Introduction and no references supporting their recognition as distinct species. The manuscript should explicitly clarify that if the species status of Begomovirus spathoglottis 1 and 2 is not determined, these viruses do not represent officially recognized species. The proposed names "Maldovirus spathoglottis 1 and 2" do not constitute a formal species description; accordingly, they should not be italicized and should be written in lowercase, as italics are reserved for ICTV-recognized binomial species names. The heading "“Proposed new species within the genus Maldovirus" could be misleading. It would be clearer to indicate that these are proposed virus names rather than formal species.

If biological evidence (e.g., fulfillment of Koch’s postulates) has been provided to support the designation of a new species, this should be clearly stated. Otherwise, the authors may consider either providing stronger evidence to justify species designation or clarifying that these are provisional virus names.

A few minor revisions remain, mainly related to grammar and sentence flow:

For grammatical accuracy, please revise lines 25–26 to either: "This study presents a comprehensive comparative genomic analysis and global distribution map using…." or "This study presents comprehensive comparative genomic analyses and a global distribution map using …".

Please revise lines 42-44: Furthermore, we hypothesized specific vector transmission: Opunvirus, Welwivirus, and Topilevirus are transmitted via treehopper species, whereas Citlodavirus and Eragrovirus are transmitted via leafhopper species.

Line 45: Consider changing "with" to "and" in the phrase "plant viruses with their transmission vectors..."

Line 54: "tomatoes production" should be corrected to "tomato production".

For lines 67-69, This emergence of new virus species and genera with modified virulence overcomes the host resistance gene and surveillance through the increased adaptation rate of the virus to environmental factors and susceptible host plants [15-17].

Suggested revision: The emergence of new virus species and genera with modified virulence can overcome host resistance genes and challenge surveillance due to the virus’s increased adaptation to environmental factors and susceptible host plants [15-17]”

Lines 74-75: “Their respective studies were mainly on begomoviruses and a few other genera." Can be changed to "These studies mainly focused on begomoviruses and a few other genera."

It should be noted that the phrase on lines 83–84 remains incomplete. The verb "is essential" was missed during the previous revision. The corrected sentence "Understanding the diversity and evolution of both new and previously reported species is essential to support surveillance efforts against emerging diseases threatening food security."

Lines 96-97: "The diversity of these genera varies from one to another" Can be rephrase as 'The genetic diversity among these genera varies."

Lines 103-106: Suggested revision for more fluent wording” Sequences for Mastrevirus and Begomovirus, which had a large number of available sequences, were grouped into batches for parallel computation, and sampling was performed using ≥80% pairwise similarity. Similarly, Bandoo et al. [28] adopted 80% pairwise identity when selecting members of Begomovirus.

Lines 125-126: "This is to avoid graphic clutter, which will subsequently simplify the data visualization.” Change to “This approach minimizes graphic clutter and facilitates data visualization."

Lines 139-141: "The integrated methods in RDP5 that were used to perform the preliminary scan for possible recombination in the aligned virus sequences are Chimaera [35], Bootscan [36–37], Geneconv [21], SisterScan [38], Maximum Chi Square [35, 39], and 3Seq [40]." can be rephrase as "The following integrated methods in RDP5 were used to perform a preliminary scan for possible recombination in the aligned virus sequences: Chimaera [35], Bootscan [36–37], Geneconv [21], SisterScan [38], Maximum Chi Square [35, 39], and 3Seq [40]."

Lines 147-151: "The analysis of the pairwise identity for the sampled genomes revealed the inter-genera pairwise identity across the monopartite of the entire genera of the Geminiviridae and the Begomovirus DNA-A to be ≥40%….." can be rephrase as "Analysis of pairwise identity for the sampled genomes revealed that inter-genera identity among monopartite geminiviruses and Begomovirus DNA-A is ≥40% (Fig. 1), corresponding to a maximum genetic variation of 60%. Similarly, interspecies pairwise identity across the complete genomes of Begomovirus DNA-B is ≥41% (Fig. 2), corresponding to a maximum genetic variation of 59%".

185-186: "Alternatively, few species of Begomovirus, including unclassified Begomovirus and Mastrevirus, clusters outside their respective genus members." can be rephrase as "However, a few Begomovirus species, including unclassified Begomovirus and Mastrevirus, cluster outside their respective genera."

Lines 208-209: "Interestingly, the species of Capulavirus was not among the entire viruses in the BLAST results." can be rephrase as "Interestingly, no Capulavirus species were detected in the BLAST results."

Lines 223-224: In the sentence "The virus nucleotide sequences and amino acid sequences of the coat protein have the highest percentage pairwise identity of 91.3% and 99.2%, respectively, to Corchorus yellow vein mosaic virus (CoYV) accessions KX513862 and AGG18212, respectively," the second ‘respectively’ is redundant. It can be removed:

"…..have the highest percentage pairwise identity of 91.3% and 99.2% to Corchorus yellow vein mosaic virus (CoYV) accessions KX513862 and AGG18212, respectively".

Line 228: Since the abbreviation "CoYV" has already been introduced, it is not necessary to write the full name "Corchorus yellow vein mosaic virus" again. Please correct to "CoYV".

Line 242: Importantly, CP emerged as the key genomic region distinguishing Begomovirus species from both Maldovirus members and Begomovirus spathoglottis 1 and 2 (Fig 11)." Consider revising to "The CP was identified as the primary genomic region distinguishing Begomovirus species from both Maldovirus members and begomovirus spathoglottis 1 and 2 (Fig 11)."

Lines 254-255: "Interspecies recombination was detected in higher numbers in Begomovirus; many species of this genus had recombinant sequences of other species in their genomes." Can be revised to "Interspecies recombination was more frequent in Begomovirus, with many species containing recombinant sequences derived from other species."

Line 329: The phrase "Contrarily, the CP of a few species…" should be revised. I recommend replacing "Contrarily" with "In contrast," which is more appropriate.

**Do you want your identity to be public for this peer review?** For information about this choice, including consent withdrawal, please see our Privacy Policy

Reviewer #1: No

Reviewer #2: No

---

## [Author Response · Author response to Decision Letter 2]

1 Oct 2025

Reviewer 1

Comment 1: L.52-56: Please cite the viral species responsible.

Response 1: We appreciate this comment, and the responsible species have been cited accordingly.

Change Made: Described in lines 52-60

Comment 2: L.107: The link https://doi.org/10.17504/protocols.io.j8nlkybjdg5r/v1 does not work.

Response 2: This observation is well appreciated. Truly, this is a reserve DOI because the protocol has not been published. The protocol publication is delayed to allow both editor and reviewers to review it, since the protocol is an integral part of this manuscript. As stated in the journal guideline, this reserve DOI will be automatically made public once the manuscript is published. As earlier included in the previous revision, this is the private link to access the protocol: https://www.protocols.io/private/9889AFB557F711F0B1B50A58A9FEAC02

Comment 3: L.169-170: I suggest changing the sentence to:

"Additionally, the African continent had the highest number of countries where geminivirus infections are reported." Instead of: "Additionally, the African continent had the highest number of countries affected with geminivirus infections."

Response 3: This observation of avoiding potential bias is well appreciated and implemented accordingly.

Change Made: Described in lines 169-170

Reviewer 2

Comment 1: The authors may propose names for new virus isolates, but it is the ICTV’s role to determine whether a new species should be formally recognized. Begomovirus spathoglottis 1 and 2 are introduced only in the Abstract and Results, with no prior mention in the Introduction and no references supporting their recognition as distinct species. The manuscript should explicitly clarify that if the species status of Begomovirus spathoglottis 1 and 2 is not determined, these viruses do not represent officially recognized species. The proposed names "Maldovirus spathoglottis 1 and 2" do not constitute a formal species description; accordingly, they should not be italicized and should be written in lowercase, as italics are reserved for ICTV-recognized binomial species names. The heading "“Proposed new species within the genus Maldovirus" could be misleading. It would be clearer to indicate that these are proposed virus names rather than formal species. If biological evidence (e.g., fulfillment of Koch’s postulates) has been provided to support the designation of a new species, this should be clearly stated. Otherwise, the authors may consider either providing stronger evidence to justify species designation or clarifying that these are provisional virus names.

Response 1: Thank you for this thoughtful comment regarding the potentially misleading subtitle related to the proposed virus names for unclassified viruses. We have clarified that these proposed names have not yet been formally classified into species and that the formal recognition of new species is the responsibility of the ICTV. The suggested convention of using lowercase for these names has been upheld.

Change Made: Described in lines 36-38 and 230-250

Comment 2: For grammatical accuracy, please revise lines 25–26 to either: "This study presents a comprehensive comparative genomic analysis and global distribution map using…." or "This study presents comprehensive comparative genomic analyses and a global distribution map using …".

Response 2: This suggestion is well appreciated and updated accordingly.

Change Made: Described in lines 24-25

Comment 3: Please revise lines 42-44: Furthermore, we hypothesized specific vector transmission: Opunvirus, Welwivirus, and Topilevirus are transmitted via treehopper species, whereas Citlodavirus and Eragrovirus are transmitted via leafhopper species.

Response 3: This suggestion is well appreciated and updated accordingly.

Change Made: Described in lines 42-44

Comment 4: Line 45: Consider changing "with" to "and" in the phrase "plant viruses with their transmission vectors..."

Response 4: This suggestion is well appreciated and updated accordingly.

Change Made: Described in line 45

Comment 5: Line 54: "tomatoes production" should be corrected to "tomato production".

Response 5: This suggestion is well appreciated and updated accordingly.

Change Made: Described in line 57

Comment 6: For lines 67-69, This emergence of new virus species and genera with modified virulence overcomes the host resistance gene and surveillance through the increased adaptation rate of the virus to environmental factors and susceptible host plants [15-17].

Suggested revision: The emergence of new virus species and genera with modified virulence can overcome host resistance genes and challenge surveillance due to the virus’s increased adaptation to environmental factors and susceptible host plants [15-17]”

Response 6: This suggestion is well appreciated and updated accordingly.

Change Made: Described in lines 70-72

Comment 7: Lines 74-75: “Their respective studies were mainly on begomoviruses and a few other genera." Can be changed to "These studies mainly focused on begomoviruses and a few other genera."

Response 7: This suggestion is well appreciated and updated accordingly.

Change Made: Described in lines 77-78

Comment 8: It should be noted that the phrase on lines 83–84 remains incomplete. The verb "is essential" was missed during the previous revision. The corrected sentence "Understanding the diversity and evolution of both new and previously reported species is essential to support surveillance efforts against emerging diseases threatening food security."

Response 8: This suggestion is well appreciated and updated accordingly.

Change Made: Described in lines 86-88

Comment 9: Lines 96-97: "The diversity of these genera varies from one to another" Can be rephrase as 'The genetic diversity among these genera varies."

Response 9: This suggestion is well appreciated and updated accordingly.

Change Made: Described in lines 99-100

Comment 10: Lines 103-106: Suggested revision for more fluent wording” Sequences for Mastrevirus and Begomovirus, which had a large number of available sequences, were grouped into batches for parallel computation, and sampling was performed using ≥80% pairwise similarity. Similarly, Bandoo et al. [28] adopted 80% pairwise identity when selecting members of Begomovirus.

Response 10: This suggestion is well appreciated and updated accordingly.

Change Made: Described in lines 106-109

Comment 11: Lines 125-126: "This is to avoid graphic clutter, which will subsequently simplify the data visualization.” Change to “This approach minimizes graphic clutter and facilitates data visualization."

Response 11: This suggestion is well appreciated and updated accordingly.

Change Made: Described in lines 128-129

Comment 12: Lines 139-141: "The integrated methods in RDP5 that were used to perform the preliminary scan for possible recombination in the aligned virus sequences are Chimaera [35], Bootscan [36–37], Geneconv [21], SisterScan [38], Maximum Chi Square [35, 39], and 3Seq [40]." can be rephrase as "The following integrated methods in RDP5 were used to perform a preliminary scan for possible recombination in the aligned virus sequences: Chimaera [35], Bootscan [36–37], Geneconv [21], SisterScan [38], Maximum Chi Square [35, 39], and 3Seq [40]."

Response 12: This suggestion is well appreciated and updated accordingly.

Change Made: Described in lines 141-143

Comment 13: Lines 147-151: "The analysis of the pairwise identity for the sampled genomes revealed the inter-genera pairwise identity across the monopartite of the entire genera of the Geminiviridae and the Begomovirus DNA-A to be ≥40%….." can be rephrase as "Analysis of pairwise identity for the sampled genomes revealed that inter-genera identity among monopartite geminiviruses and Begomovirus DNA-A is ≥40% (Fig. 1), corresponding to a maximum genetic variation of 60%. Similarly, interspecies pairwise identity across the complete genomes of Begomovirus DNA-B is ≥41% (Fig. 2), corresponding to a maximum genetic variation of 59%".

Response 13: This suggestion is well appreciated and updated accordingly.

Change Made: Described in lines 149-152

Comment 14: 185-186: "Alternatively, few species of Begomovirus, including unclassified Begomovirus and Mastrevirus, clusters outside their respective genus members." can be rephrase as "However, a few Begomovirus species, including unclassified Begomovirus and Mastrevirus, cluster outside their respective genera."

Response 14: This suggestion is well appreciated and updated accordingly.

Change Made: Described in lines 184-185

Comment 15: Lines 208-209: "Interestingly, the species of Capulavirus was not among the entire viruses in the BLAST results." can be rephrase as "Interestingly, no Capulavirus species were detected in the BLAST results."

Response 15: This suggestion is well appreciated and updated accordingly.

Change Made: Described in lines 208-209

Comment 16: Lines 223-224: In the sentence "The virus nucleotide sequences and amino acid sequences of the coat protein have the highest percentage pairwise identity of 91.3% and 99.2%, respectively, to Corchorus yellow vein mosaic virus (CoYV) accessions KX513862 and AGG18212, respectively," the second ‘respectively’ is redundant. It can be removed:

"…..have the highest percentage pairwise identity of 91.3% and 99.2% to Corchorus yellow vein mosaic virus (CoYV) accessions KX513862 and AGG18212, respectively".

Response 16: This suggestion is well appreciated and updated accordingly.

Change Made: Described in lines 223-225

Comment 17: Line 228: Since the abbreviation "CoYV" has already been introduced, it is not necessary to write the full name "Corchorus yellow vein mosaic virus" again. Please correct to "CoYV".

Response 17: This suggestion is well appreciated and updated accordingly.

Change Made: Described in line 228

Comment 18: Line 242: Importantly, CP emerged as the key genomic region distinguishing Begomovirus species from both Maldovirus members and Begomovirus spathoglottis 1 and 2 (Fig 11)." Consider revising to "The CP was identified as the primary genomic region distinguishing Begomovirus species from both Maldovirus members and begomovirus spathoglottis 1 and 2 (Fig 11)."

Response 18: This suggestion is well appreciated and updated accordingly.

Change Made: Described in lines 242-243

Comment 19: Lines 254-255: "Interspecies recombination was detected in higher numbers in Begomovirus; many species of this genus had recombinant sequences of other species in their genomes." Can be revised to "Interspecies recombination was more frequent in Begomovirus, with many species containing recombinant sequences derived from other species."

Response 19: This suggestion is well appreciated and updated accordingly.

Change Made: Described in lines 255-256

Comment 20: Line 329: The phrase "Contrarily, the CP of a few species…" should be revised. I recommend replacing "Contrarily" with "In contrast," which is more appropriate.

Response 20: This suggestion is well appreciated and updated accordingly.

Change Made: Described in line 330

---

## [Decision Letter · Decision Letter 2]

25 Nov 2025

Diversity, recombination and misclassification in the family Geminiviridae: Insight from bioinformatics analysis

PONE-D-25-15367R2

Dear Dr. Bretana,

We’re pleased to inform you that your manuscript has been judged scientifically suitable for publication and will be formally accepted for publication once it meets all outstanding technical requirements.

Kind regards,

Shunmugiah Veluchamy Ramesh, PhD

Academic Editor

PLOS ONE

Additional Editor Comments (optional):

Address these minor issues:

In lines 53–55, the newly added text containing virus names is again capitalized and italicized, despite my previous comments requesting that common virus names be written in lowercase and without italics.

Please correct these and also provide the corresponding ICTV-approved binomial species names (italicized) after each virus name. For example:

“Notably, cotton leaf curl disease (CLCuD), primarily caused by Begomovirus gossypimultanense and Begomovirus gossypikokranense in Pakistan and India [7], has hindered cotton production in these countries, while African cassava mosaic virus (ACMV, Begomovirus manihotis) is responsible for cassava mosaic disease across Africa [9]. In the United States, tomato yellow leaf curl virus (TYLCV, Begomovirus coheni) has significantly affected…..”

Line 57: Please “CLCuV” Change to “CLCuD”

Reviewers' comments:

Reviewer's Responses to Questions

**Comments to the Author**

Reviewer #1: All comments have been addressed

Reviewer #2: All comments have been addressed

2. Is the manuscript technically sound, and do the data support the conclusions?

Reviewer #1: Yes

Reviewer #2: Yes

3. Has the statistical analysis been performed appropriately and rigorously?

Reviewer #1: Yes

Reviewer #2: Yes

4. Have the authors made all data underlying the findings in their manuscript fully available?

Reviewer #1: Yes

Reviewer #2: Yes

5. Is the manuscript presented in an intelligible fashion and written in standard English?

Reviewer #1: Yes

Reviewer #2: Yes

Reviewer #1: This is my final review of the manuscript, which presents an interesting and valuable contribution regarding the importance of regularly updating public databases such as GenBank, or at least exercising caution when using them in phylogenetic studies focused on a single genus. I appreciate the authors’ efforts to address the points I raised in my previous two reviews. The manuscript is well written and provides a thorough assessment of the genetic diversity, phylogeny, and global distribution of the Geminiviridae family. Overall, given the completeness of their responses to my comments and those of the other reviewers, I recommend the manuscript for acceptance.

Reviewer #2: The authors have adequately addressed the reviewers’ requested revisions in this round, and the manuscript has improved. However, a minor issue remains.

In lines 53–55, the newly added text containing virus names is again capitalized and italicized, despite my previous comments requesting that common virus names be written in lowercase and without italics.

Please correct these and also provide the corresponding ICTV-approved binomial species names (italicized) after each virus name. For example:

“Notably, cotton leaf curl disease (CLCuD), primarily caused by Begomovirus gossypimultanense and Begomovirus gossypikokranense in Pakistan and India [7], has hindered cotton production in these countries, while African cassava mosaic virus (ACMV, Begomovirus manihotis) is responsible for cassava mosaic disease across Africa [9]. In the United States, tomato yellow leaf curl virus (TYLCV, Begomovirus coheni) has significantly affected…..”

Line 57: Please “CLCuV” Change to “CLCuD”

The authors are requested to address these issues in the manuscript.

**Do you want your identity to be public for this peer review?** For information about this choice, including consent withdrawal, please see our Privacy Policy

Reviewer #1: No

Reviewer #2: No

---

## [Editor Report · Acceptance letter]

PONE-D-25-15367R2

PLOS One

Dear Dr. Bretana,

I'm pleased to inform you that your manuscript has been deemed suitable for publication in PLOS One. Congratulations! Your manuscript is now being handed over to our production team.

Kind regards,

on behalf of

Dr. Shunmugiah Veluchamy Ramesh

Academic Editor

PLOS One